# Recruitment of BRD4 to the ASXL1 genomic targets depends on the extra-terminal domain of BRD4

Karthik Selvam [1], Shuang Lu[2,3], Clémence Messmer[4], Yakun Pang[5], Soumi Biswas[1], Moustafa Khalil[4], Peng Zhang [5], Rima Tulaiha[6], Ming-Ming Zhou [7], Toshio Kitamura [8], Shannon M. Lauberth [6], M. Andres Blanco [9], Feng-Chun Yang [5], El Bachir Affar [4,10], Zibo Zhao [6], Lei Zeng [2,3] ✉, Lu Wang [6] ✉ & Tatiana G. Kutateladze [1] ✉

ASXL1 is a well-established driver of a wide range of cancers. Here, we identify a high level of genetic correlation between ASXL1 and the major transcriptional activator BRD4 in cancer cells and characterize the molecular mechanism underlying this correlation. Structural and biochemical data show the formation of a tight complex between the extraterminal domain of BRD4 and the BRD4-binding motif of ASXL1. ChIP-seq analysis of mutated ASXL1 that cannot bind BRD4 demonstrates that the recruitment of BRD4 to the ASXL1 genomic targets depends on this interaction. We find that the cancer-related truncated variants comprising residues 1-645 and 1-591 of ASXL1 (ASXL1$^{1\text{-}645}$ and ASXL1$^{1\text{-}591}$) retain BRD4 binding function, with ASXL1$^{1\text{-}645}$ showing an enhanced ability to recruit BRD4 to promoters of ASXL1 target genes. In contrast to ASXL1$^{1\text{-}591}$, ASXL1$^{1\text{-}645}$ simultaneously and independently interacts with BRD4 and the H3K4-specific methyltransferases MLL3/4, whereas the shorter variant is lacking the MLL3/4-binding site and interacts only with BRD4. Genomic data from six cancer types reveals a strong positive ASXL1-BRD4 relationship, with BRD4 occupying the ASXL1 promoter and thus pointing to a possible feed-forward mechanism. Our findings provide mechanistic details by which ASXL1 associates with BRD4 and shed light on the biological significance of this association.

The transcriptional coregulator and a core component of the Polycomb Repressive Deubiquitinase (PR-DUB) complex, ASXL1 (additional sex combs like 1), is frequently mutated and truncated in human malignances, particularly in acute and chronic myeloid leukemias (AML and CML, respectively) and myelodysplastic syndromes (MDS)[1–7] (Fig. 1a). ASXL1 stimulates activity of the catalytic subunit of the PR-DUB complex, the ubiquitin hydrolase BAP1, that deubiquitinates mono-ubiquitinated lysine 119 of histone H2A, a mark of repressive chromatin deposited by the Polycomb repressive complex-1 (PRC1) to maintain Polycomb group (PcG) target gene silencing and preserve

cellular transcriptional identity[5,8,9]. Additionally, ASXL1 was reported to associate with components of the H3K27me3-specific PRC2 complex, nuclear hormone receptors, BRD4 and MLL3/4 and therefore functions as a context-specific gene regulator[7,10]. ASXL1 and its homologs ASXL2 and ASXL3 share a common domain architecture of a HARE-HTH domain, the BAP1-binding DEUBAD domain, the MLL3/4 binding helix (MBH) and a C-terminal zinc finger (Fig. 1b).

Like ASXL1, the transcriptional co-activator bromodomain and extra-terminal domain 4 (BRD4) is also linked to tumorigenesis. It forms chromosomal translocations in squamous carcinoma and NUT

midline carcinoma, plays a role in the progression of AML, and is upregulated in breast cancer[11–17]. BRD4 contains two acetyllysine-binding bromodomains with distinct functions (Fig. 1b). The first bromodomain (BD1) binds to poly-acetylated histones, such as H4K5acK8ac, whereas the second bromodomain (BD2) selects for acetylated non-histone proteins[18–20]. The extra-terminal (ET) domain of BRD4 recognizes the hKhKh motif (where $h$ is a hydrophobic residue) found in LANA, CHD4, BRG1, integrase, and NSDs[21–23]. Functional domains of BRD4 are essential in the recruitment and/or stabilization of BRD4-containing transcription complexes at target genes and stimulation of transcription[23–25]. A region in ASXL1 has recently been identified as a binding partner of the ET domain of BRD4[26], prompting us to explore the mechanistic details and the functional significance of this interaction.

In this study, we report that ASXL1, BRD4 and BAP1 colocalize at promoters of transcriptionally active genes with ASXL1 linking BRD4 and BAP1. We describe the molecular and structural basis for the recognition of ASXL1 and homologous ASXL2 and ASXL3 by the ET domain of BRD4 (BRD4$_{ET}$). Our findings underscore the critical role of the BRD4-ASXL1 interaction in bridging BRD4 to the ASXL1-bound genomic sites and reveal a high level of correlation between ASXL1 and BRD4 in malignancies.

## Results and Discussion

### BRD4$_{ET}$ binds to ASXL1$_{EBM}$

Binding of BRD4$_{ET}$ to an ASXL1 peptide was originally observed in biochemical pulldown experiments[26]. To assess whether this interaction is physiologically relevant, we immunoprecipitated ASXL1 from soluble nuclear extracts of K562 cells expressing Flag-Strep-tagged ASXL1 and identified potential ASXL1-binding partners by mass spectrometry (MS). As anticipated, MS analysis showed co-fractionation of Flag-Strep ASXL1 with other subunits of the PR-DUB BAP1 complex, including BAP1, HCF-1 and FOXK1/2, as well as MLL3 from MLL3/4 complexes (Fig. 1c). High spectral counts were also observed for BRD4 and BRD3, suggesting that ASXL1 and BRD4/3 form complexes in the cell.

To further explore the association of ASXL1 and BRD4 in the cellular context, we performed chromatin immunoprecipitation coupled with deep sequencing (ChIP-seq) of endogenous ASXL1, BRD4 and BAP1 and assessed genomic occupancies of these proteins in human HEK293T cells. We also analyzed ChIP-seq datasets of histone H3 modifications, including H3K4me3, H3K4me1, H3K27ac, and H3K27me3, previously generated by us from human HEK293T cells[27]. As shown in Fig. 1d, a large set of the BRD4 chromatin binding sites were also co-bound by ASXL1 and BAP1. ASXL1 peaks were primarily promoter localized ( ~ 70%), and BRD4 and ASXL1 co-bound genomic sites had substantial overlap with the marks of active promoters, H3K4me3 and H3K27ac (Fig. 1d, e). In contrast, no BRD4 and ASXL1 co-occupancy was observed on H3K4me1-rich enhancers or H3K27me3-rich transcriptionally silenced regions (Fig. 1d). Together, these results suggested a potential cooperativity of BRD4 and ASXL1 that co-occupy transcriptionally active promoters.

We then used NMR and isothermal titration calorimetry (ITC) assays to validate the direct association of BRD4 with ASXL1. We produced $^{15}$N-labeled BRD4$_{ET}$ and recorded its $^1$H,$^{15}$N heteronuclear single quantum coherence (HSQC) spectra while unlabeled ASXL1 peptide (aa 568–581 of ASXL1) was gradually added to the NMR sample. The ASXL1 peptide induced large chemical shift perturbations (CSPs) in BRD4$_{ET}$, indicative of the binding (Fig. 1f). Many amide resonances of BRD4$_{ET}$ in the apo-state disappeared upon addition of the peptide (from here on referred to as the ET-binding motif of ASXL1 (ASXL1$_{EBM}$)), and another set of resonances corresponding to the ASXL1-bound state of BRD4$_{ET}$ appeared. The slow exchange regime on the NMR timescale suggested a tight binding, which was confirmed by measuring the binding affinity of BRD4$_{ET}$ to ASXL1$_{EBM}$ by ITC. The

dissociation constant ($K_d$) was found to be 1.6 μM, and was in agreement with the previously reported affinity of 1.2 μM[26] (Fig. 1g).

### Molecular mechanism for the recognition of ASXL1 by BRD4$_{ET}$

We note that the ASXL1$_{EBM}$ peptide contains an atypical (hRhQh) sequence, which is divergent from the canonical ET-binding motif. To gain insight into the binding mechanism, we determined the three-dimensional solution structure of BRD4$_{ET}$ in complex with ASXL1$_{EBM}$ using NMR spectroscopy. In the complex, BRD4$_{ET}$ folds into a three-helix bundle, observed in other ET domain structures[21–23,28], whereas ASXL1$_{EBM}$ is bound in an extended conformation and makes extensive intermolecular contacts with BRD4$_{ET}$ (Fig. 2 and Supplementary Fig. 1). ASXL1$_{EBM}$ pairs with the loop connecting α2 and α3 of BRD4$_{ET}$, resulting in the formation of the double-stranded anti-parallel β-sheet (Fig. 2a). Overall, the elongated ASXL1$_{EBM}$-binding groove, delineated by I622, L630, V633, V634, I652, I654 and F656 of BRD4$_{ET}$, is mainly hydrophobic but is surrounded by the negatively charged walls created by the residues of the β1 strand and the α2-α3 loop of BRD4$_{ET}$, such as D650, E651, E653, D655 and E657 (Fig. 2b, c). Short distances, less than 3.5 Å, observed between the backbone amides and the carbonyl groups of I574 and I572 of ASXL1$_{EBM}$ and the backbone carbonyl groups and amides of I652 and I654 of BRD4$_{ET}$, indicate the formation of the canonical β sheet contacts (Fig. 2c). The side chains of V569, I572, I574 and L576 of ASXL1$_{EBM}$ are oriented toward and essentially buried in the hydrophobic groove of BRD4$_{ET}$. The complex is further stabilized by a set of apparently hydrogen bonds formed between the backbone amide of V569 of ASXL1$_{EBM}$ and the side chain carboxyl group of E657 of BRD4$_{ET}$ and between the backbone amide of L576 of ASXL1$_{EBM}$ and the backbone carbonyl of D650 of BRD4$_{ET}$. Several electrostatically driven side-chain to side-chain contacts are observed, particularly involving the carboxyl groups of D650, E653, and E657 of BRD4$_{ET}$ and the guanidino and ammonium moieties of R578, R573, and K568 of ASXL1$_{EBM}$, respectively, and the carboxyl group of E651 of BRD4$_{ET}$ is likely engaged with the side chain amide nitrogen of Q575 of ASXL1$_{EBM}$.

### Binding of BRD4$_{ET}$ to ASXLs is conserved

ASXL1 and homologous ASXL2 and ASXL3 were identified as mutually exclusive components of the PR-DUB complex[1,29], and all contain similar but not identical ET-binding sequences (IRIQL in ASXL1, LKIPV in ASXL2 and LKIQL in ASXL3) (Supplementary Fig. 2). To determine whether the ASXLs binding mechanism is conserved in BRD4$_{ET}$, we tested ASXL2$_{EBM}$ (aa 615-628 of ASXL2) and ASXL3$_{EBM}$ (aa 1008–1021 of ASXL3) peptides in $^1$H,$^{15}$N HSQC titration experiments and fluorescence spectroscopy (Fig. 3). Addition of ASXL2$_{EBM}$ or ASXL3$_{EBM}$ caused large CSPs in $^{15}$N-labeled BRD4$_{ET}$ that were on par with CSPs induced in BRD4$_{ET}$ by ASXL1$_{EBM}$ (Fig. 3a, b). Although not strictly identical, similar patterns of CSPs observed in BRD4$_{ET}$ per residue upon titration with either ASXLs suggested that all ASXLs can associate with BRD4$_{ET}$, and each occupies the same binding site of BRD4$_{ET}$ (Fig. 3e–g). In further support, $K_d$s of 1–2 μM were measured for the interaction of BRD4$_{ET}$ with ASXL1$_{EBM}$, ASXL2$_{EBM}$ and ASXL3$_{EBM}$ peptides in fluorescence assays (Fig. 3c, d). Collectively, these data indicated that the ASXL1/2/3 binding mode of BRD4$_{ET}$ is conserved.

### Mapping the BRD4-ASXL1 interface

To assess the importance of the interface residues in the BRD4$_{ET}$-ASXL1$_{EBM}$ complex formation, we substituted R573 and I574 in ASXL1$_{EBM}$ individually to an aspartate and tested R573D and I574D mutants in $^1$H,$^{15}$N HSQC titration experiments and fluorescence assays (Fig. 4a, b and Supplementary Fig. 3). These residues were selected because the hydrophobic side chain of I574 inserts into a relatively deep and narrow pocket of BRD4$_{ET}$, and the guanidino moiety of R573 is restrained through the contact with the carboxyl group of E653 of BRD4$_{ET}$. As shown in Fig. 4a, mutation of I574 in ASXL1$_{EBM}$ essentially

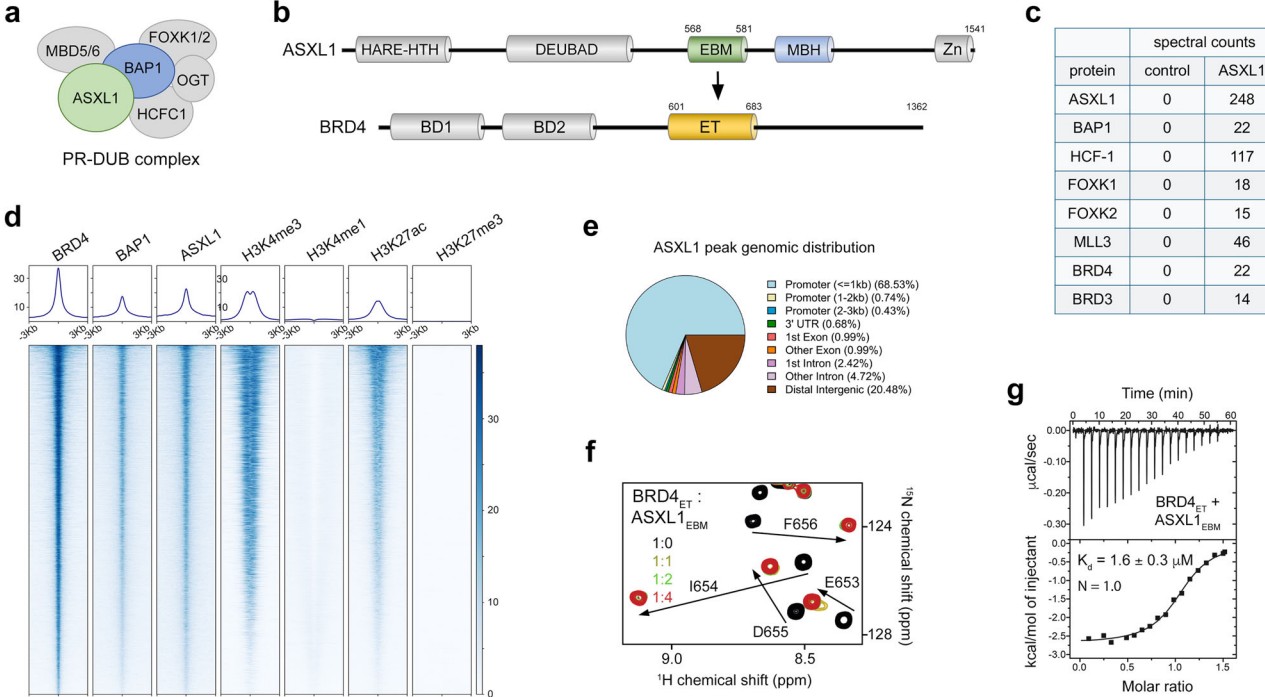

**Fig. 1 | BRD4$_{ET}$ binds to ASXL1$_{EBM}$. a** Schematic of the human Polycomb Repressive Deubiquitinase (PR-DUB) BAP1 complex, consisting of the indicated subunits. **b** ASXL1 and BRD4 domain architecture. **c** Proteins identified by mass spectrometry proteomic analysis of purified fractions from K562 cells expressing Flag-Strep-ASXL1. Mock, control. **d** Heat maps of ChIP-seq signals of the indicated endogenous full length proteins and PTMs at genomic regions bound by BRD4 in HEK293T cells.

**e** ASXL1 peak genomic distribution. **f** Overlaid $^1$H,$^{15}$N HSQC spectra of BRD4$_{ET}$ collected upon titration with ASXL1$_{EBM}$ (aa 568-581 of ASXL1) peptide. Spectra are color-coded according to the protein:peptide molar ratio. **g** ITC titration and fitting curve used to determine the binding affinity of BRD4$_{ET}$ to ASXL1$_{EBM}$ peptide. K$_d$ represents the average of three independent measurements ± SD.

abolished binding to BRD4$_{ET}$ and indicated that the hydrophobic contacts are the major driving force of this interaction. Binding activity of the R573D mutant of ASXL1$_{EBM}$ was substantially decreased (Fig. 4b and Supplementary Fig. 3), and a comparable decrease was observed in the reciprocal binding of wild type ASXL1$_{EBM}$ to the E653K mutant of BRD4$_{ET}$ or the E651K mutant impaired in the contact with Q575 of ASXL1$_{EBM}$ (Supplementary Fig. 3). The double E651K/E653K mutant of BRD4$_{ET}$ associated with ASXL1$_{EBM}$ very weakly (K$_d$ of >1 mM) (Fig. 4c and Supplementary Fig. 3), corroborating the structural data that electrostatic and polar interactions are also essential.

### Independent binding of BRD4$_{ET}$ to ASXL1$_{EBM}$ and DNA

Analysis of the electrostatic surface potential of the BRD4$_{ET}$-ASXL1$_{EBM}$ complex revealed that the side of BRD4$_{ET}$ which is opposite to the ASXL1$_{EBM}$ binding groove is highly positively charged (Fig. 4d, e). Particularly, α3 and the loop following α3 contain the positively charged surface residues R665, K669, R676, K677, K678, R679 and K680 that could potentially interact with negatively charged DNA. To determine whether BRD4$_{ET}$ is capable of binding to DNA, we mutated these arginine and lysine residues and examined the association of wild-type BRD4$_{ET}$ and R665E, K669E, and R676A/K677A/K678A/R679A/K680A mutants of BRD4$_{ET}$ with 147 bp 601 DNA in an electrophoretic mobility shift assay (EMSA) (Fig. 4f, g). Increasing amounts of wild-type or mutated BRD4$_{ET}$ were incubated with 601 DNA, and the reaction mixtures were resolved on a native polyacrylamide gel. A gradual increase in wild type BRD4$_{ET}$ concentration caused a shift of the DNA band, indicating formation of the DNA-BRD4$_{ET}$ complex (Fig. 4f). The R665E, K669E and R676A/K677A/K678A/R679A/K680A mutants of BRD4$_{ET}$ however lost their ability to bind DNA (Fig. 4g). We concluded that the positively charged region of BRD4$_{ET}$, encompassing α3 and the loop following α3 and located on the side opposed to

the ASXL1$_{EBM}$ binding site of BRD4$_{ET}$, is involved in the interaction with DNA. The association of wild-type BRD4$_{ET}$ with DNA was largely unaffected in the presence of ASXL1$_{EBM}$ in EMSA, confirming that the two binding sites located on the opposite sides of BRD4$_{ET}$ do not overlap (Fig. 4f). In further support, binding affinity of BRD4$_{ET}$ to ASXL1$_{EBM}$ in the presence of DNA remained unchanged (Supplementary Fig. 4).

Bromodomains (BDs) of BRD4 have previously been shown to interact with RNAs to promote transcriptional activation[30]. We generated full-length BRD4 lacking both bromodomains (ΔBDs, aa 58-169 and 349-461, referred to as FL BRD4$_{ΔBDs}$), and tested its binding to the −490 *MYC* eRNA and *PNP* mRNA. As shown in Fig. 4h, i, the native BRD4 isoform (BRD4$_{1-722}$), which contains bromodomains and BRD4$_{ET}$ bound robustly to both RNAs; however, deletion of BDs eliminated these interactions, indicating that BRD4$_{ET}$ does not appreciably contribute to the association of BRD4 with the enhancer-derived or coding RNAs.

### ASXL1$_{EBM}$-dependent recruitment of BRD4 to ASXL1 targets

Frameshift variants of ASXL1, including truncated mutants comprising amino acids 1-591 and 1-645 (ASXL1$^{1-591}$ and ASXL1$^{1-645}$), have been linked to various cancers, most frequently to myeloid malignancies and clonal hematopoiesis[31–35] (Fig. 5a). Both pathological variants contain the K568-R578 sequence which is necessary for binding of BRD4$_{ET}$ (Fig. 2c), and both variants expressed as GFP fusion proteins, GFP-ASXL1$^{1-591}$ and GFP-ASXL1$^{1-645}$, immunoprecipitated endogenous BRD4 in HEK293T cells (Fig. 5b and Supplementary Fig. 5). The I574D mutation in ASXL1 that eliminated binding of BRD4$_{ET}$ in biochemical assays (Fig. 4a), also abrogated interactions of GFP-ASXL1$^{1-591}$-I574D and GFP-ASXL1$^{1-645}$-I574D mutants with endogenous BRD4 (Fig. 5b). These data indicated that the disease-associated truncated variants can be used to

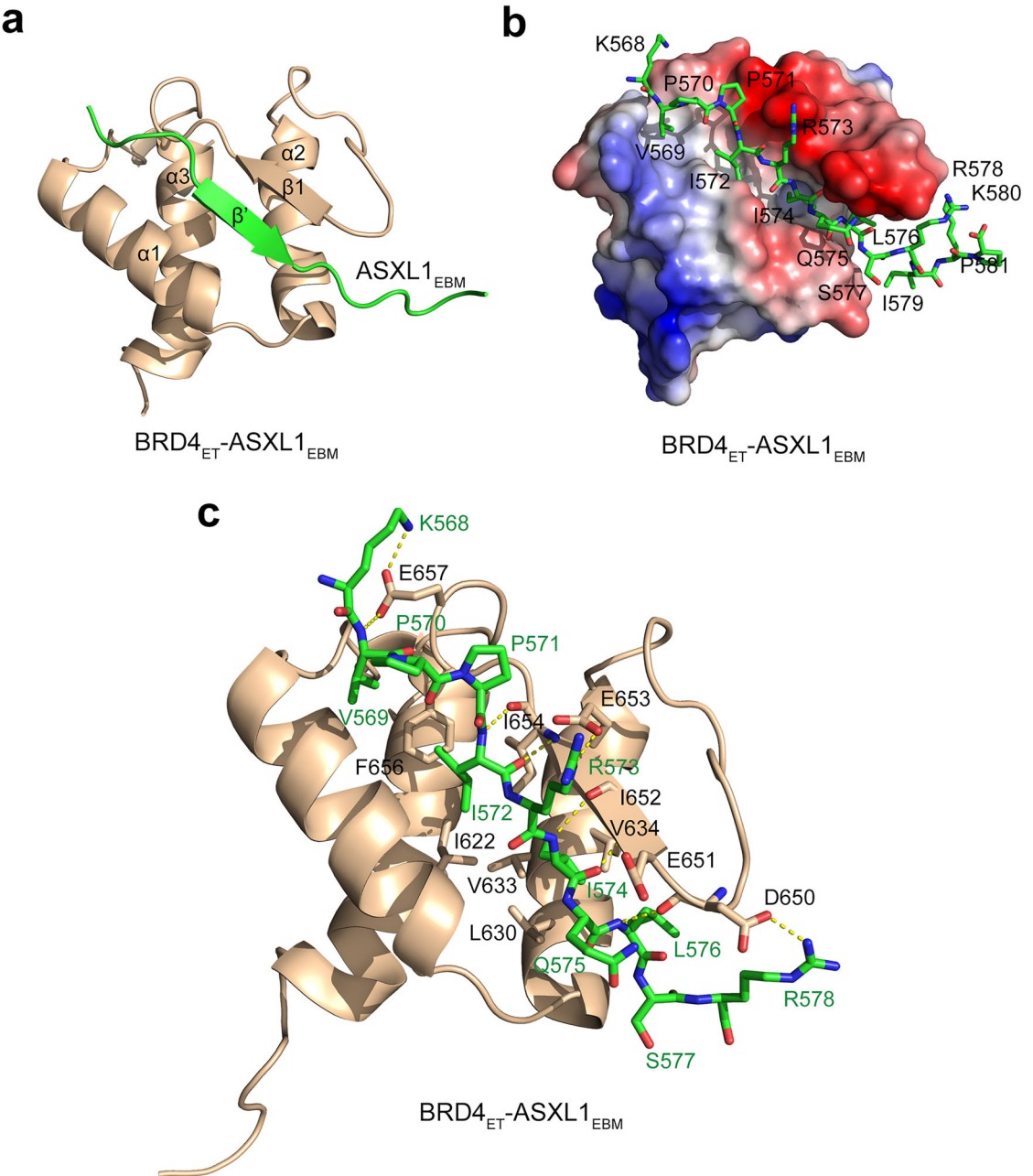

**Fig. 2 | Structure of BRD4$_{ET}$-ASXL1$_{EBM}$ complex. a** A ribbon diagram of the solution NMR structure of the BRD4$_{ET}$ (wheat) in complex with ASXL1$_{EBM}$ (green). **b** Electrostatic surface potential of the ASXL1$_{EBM}$-bound BRD4$_{ET}$ colored blue and red for positive and negative charges, respectively. ASXL1$_{EBM}$ residues are shown in green sticks and labeled. **c** Ribbon diagram of the BRD4$_{ET}$-ASXL1$_{EBM}$ complex. Residues involved in the interaction between BRD4$_{ET}$ and ASXL1$_{EBM}$ are shown as sticks labeled. Dashed lines represent short distances (less than 3.5 Å).

validate the functional significance of the association of BRD4 with ASXL1 in vivo.

ChIP-seq analysis of HEK293T cells stably expressing wild-type GFP-ASXL1$^{1-645}$ (ASXL1$^{1-645}$-WT) showed that endogenous BRD4 colocalized with ASXL1 at the transcription start sites (TSS) when signals were centered on ASXL1 peaks in ASXL1$^{1-645}$-WT (Fig. 5c, d). The positive correlation of endogenous BRD4 and ASXL1$^{1-645}$-WT, endogenous full length ASXL1 (ASXL1 FL) and BAP1 levels at TSS was also observed at individual genes, such as *ECH1* and *POGLUT1* (Fig. 5e). However, BRD4 recruitment to the ASXL1 bound TSS was reduced in GFP-ASXL1$^{1-645}$-I574D expressing cells, indicating that BRD4$_{ET}$ is necessary for the recruitment of BRD4 to the ASXL1 target genes (Fig. 5c–e). Gene Ontology (GO) enrichment analysis revealed a high degree of colocalization of ASXL1 and BRD4 on the genes associated with the regulation

of RNAs and ribosome functions (Fig. 5f), which was in line with the previously reported roles of BRD4[30].

ChIP-seq analysis at individual genes also showed that the ASXL1 and BRD4 co-bound promoters are enriched in the active promoter marks H3K4me3 and H3K27ac but depleted of H3K4me1 and H3K27me3 (Fig. 6a), suggesting a potential transcriptional activation function of the ASXL1/BRD4/BAP1 axis. ChIP-qPCR assays performed using an anti GFP antibody in HEK293T cells transduced with GFP-ASXL1$^{1-645}$-WT or GFP-ASXL1$^{1-645}$-I574D as well as with GFP-ASXL1$^{1-591}$-WT or GFP-ASXL1$^{1-591}$-I574D showed that both truncated variants of ASXL1 localize to the promoters of *ECH1*, *POGLUT1* and *PGK1* independently of binding of BRD4 (Fig. 6b). As BRD4 is an established co-activator, we next evaluated BRD4 levels at the *PGK1* gene that encodes a glycolytic enzyme known to be upregulated in glioblastoma[36] and the BRD4 target gene *VIM*[37]. ChIP-qPCR

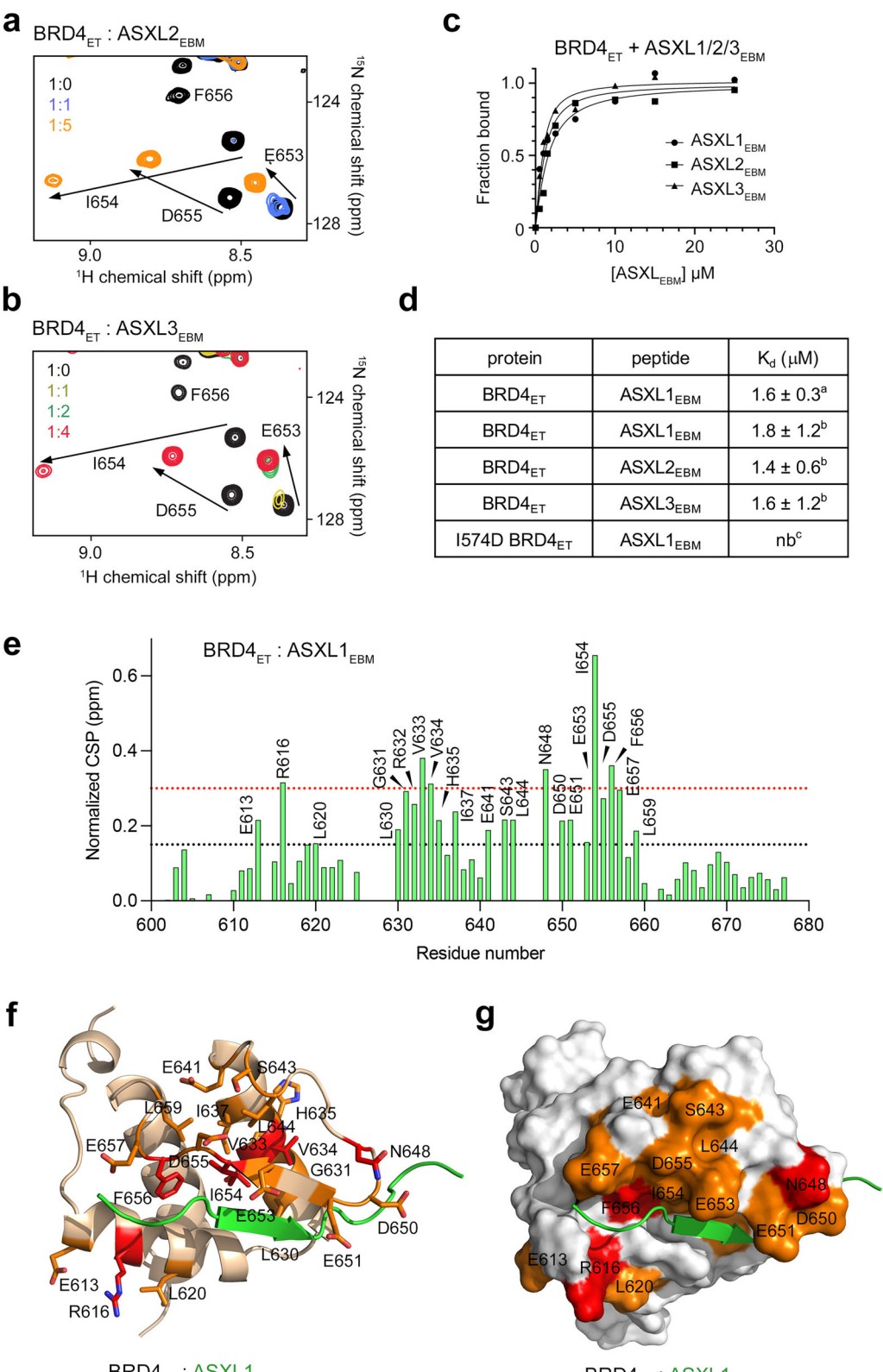

**Fig. 3 | Binding of BRD4$_{ET}$ to ASXLs is conserved. a, b** Overlaid $^{1}$H,$^{15}$N HSQC spectra of BRD4$_{ET}$ collected upon titration with ASXL2$_{EBM}$ (aa 615-628 of ASXL2) and ASXL3$_{EBM}$ (aa 1008-1021 of ASXL3) peptides. Spectra are color-coded according to the protein:peptide molar ratio. **c** Binding curves used to determine binding affinities of BRD4$_{ET}$ for the ASXL1$_{EBM}$, ASXL2$_{EBM}$ and ASXL3$_{EBM}$ peptides measured by fluorescence spectroscopy. **d** Summary of binding affinities of BRD4$_{ET}$ for the indicated peptides measured by ($^{a}$) ITC, ($^{b}$) fluorescence spectroscopy or ($^{c}$) NMR. The K$_d$ values represent average of three independent measurements (four for ASXL3), and errors represent standard deviation. **e** Normalized CSPs observed in $^{1}$H,$^{15}$N HSQC spectra of BRD4$_{ET}$ in the presence of four molar equivalents of ASXL1$_{EBM}$. The dotted black and red lines define 1x and 2x significant perturbations (1/2 average + ¾ SD). **f, g** The most perturbed residues of BRD4$_{ET}$ from (**e**) are mapped onto the ribbon diagram (**f**) or surface (**g**) of the BRD4$_{ET}$-ASXL1$_{EBM}$ structure. Residues that exhibit CSPs greater than red and black dotted lines are colored red and orange, respectively, and labeled.

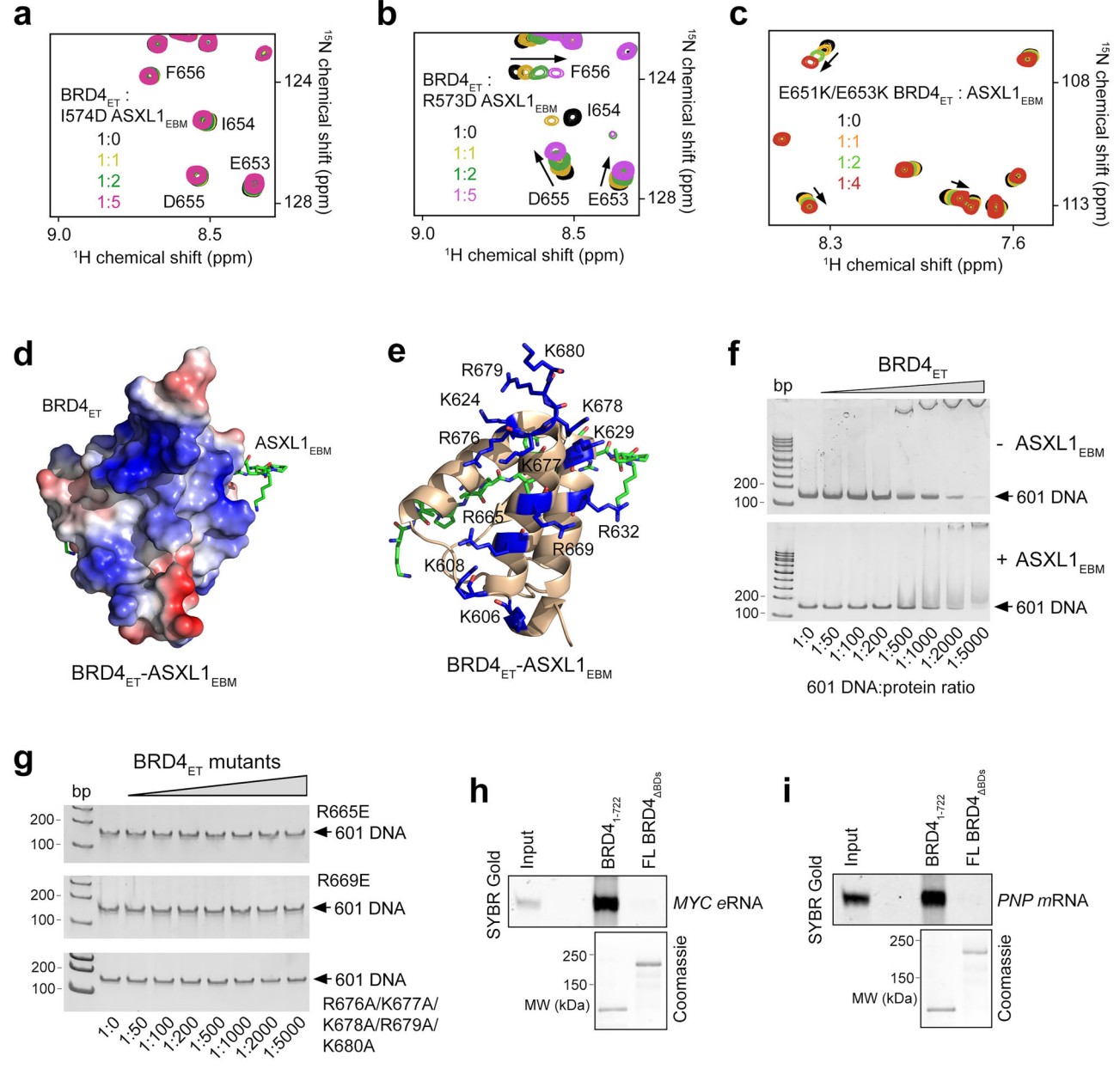

**Fig. 4 | Independent ASXL1_EBM- and DNA-binding functions of BRD4_ET.**
**a, b** Overlaid $^1$H,$^{15}$N HSQC spectra of wild-type BRD4_ET collected upon titration with the indicated mutated ASXL1_EBM peptides. Spectra are color-coded according to the protein:peptide molar ratio. **c** Overlaid $^1$H,$^{15}$N HSQC spectra of mutated BRD4_ET collected upon titration with wild-type ASXL1_EBM peptide. Spectra are color-coded according to the protein:peptide molar ratio. **d** Electrostatic surface potential of the ASXL1_EBM-bound BRD4_ET colored blue and red for positive and negative charges, respectively. ASXL1_EBM residues are shown in green sticks. **e** Ribbon diagram of the BRD4_ET-ASXL1_EBM complex. Residues of the positively charged parches in BRD4_ET are shown as blue sticks and labeled. **f** EMSAs of 601 DNA in the presence of increasing amounts of BRD4_ET with or without ASXL1_EBM. DNA:protein ratio is shown below the gel images. **g** EMSAs of 601 DNA in the presence of increasing amounts of mutated BRD4_ET. DNA:protein ratio is shown below the gel images. **h, i** In vitro pulldown of the −490 *MYC* eRNA and *PNP* mRNA with FLAG-tagged FL BRD4 DBDs. BRD4_{1-722}, positive control. SYBR Gold staining was used to detect recovered RNA, and SDS-PAGE followed by Coomassie staining was used to confirm input protein integrity. $n = 3$ independent experiments; representative images are shown in the corresponding panels. Source data are provided as a Source Data file.

assays performed using an anti-BRD4 antibody revealed a markedly higher enrichment of BRD4 at *PGK1* and *VIM* promoters in GFP-ASXL1$^{1-645}$-WT expressing cells compared to the enrichment of BRD4 in GFP-ASXL1$^{1-645}$-I574D expressing cells, indicating the ASXL1_EBM-dependent recruitment of BRD4 (Fig. 6c and Supplementary Fig. 6).

Although both GFP-ASXL1$^{1-645}$ and GFP-ASXL1$^{1-591}$ constructs contain ASXL1_EBM, occupancy of BRD4 on the *PGK1* and *VIM* promoters in GFP-ASXL1$^{1-591}$-WT expressing HEK293T cells was decreased compared to the occupancy of BRD4 on these promoters in GFP-ASXL1$^{1-645}$-WT

expressing cells (Fig. 6c and Supplementary Fig. 6). To confirm this finding, we carried out ChIP-qPCR using anti BRD4 and anti H3K4me3 antibodies in acute myeloid leukemia Kasumi-1 and chronic myelogenous leukemia K562 cell lines, characterized by the presence of endogenous ASXL1$^{1-645}$ and ASXL1$^{1-591}$ truncation variants, respectively. We observed a marked reduction in BRD4 enrichment at promoter regions of *PGK1* and *POGLUT1* and a somewhat similar level at the *ECH1* promoter in K562 cells expressing the shorter variant, ASXL1$^{1-591}$, compared to the BRD4 enrichment in Kasumi-1 cells expressing the

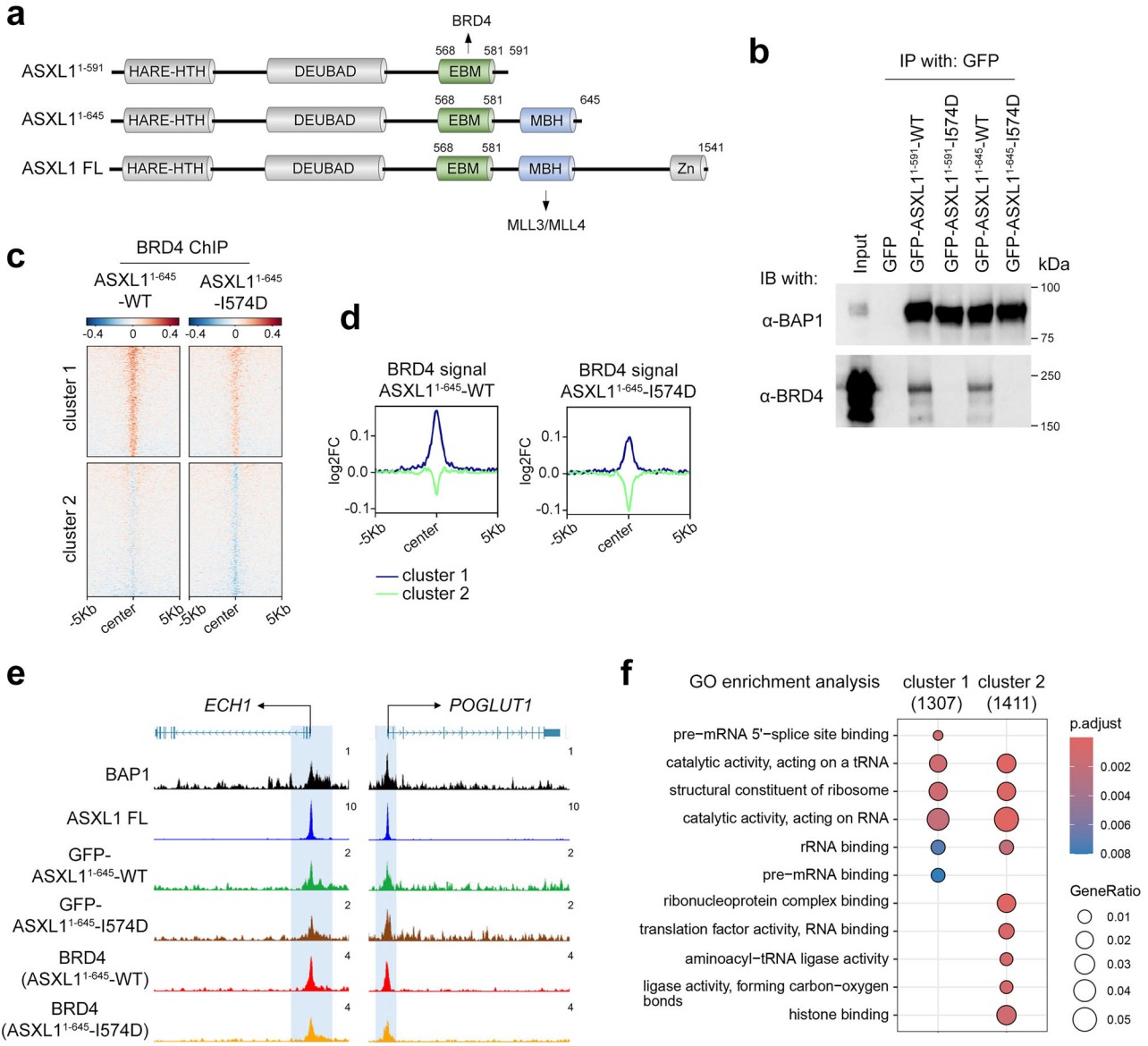

**Fig. 5 | ASXL1$_{EBM}$ is essential for the recruitment of BRD4 to the ASXL1-bound genes. a** Schematics of full-length ASXL1 and BRD4 and the indicated truncated ASXL1. **b** HEK293T cells were transduced with lenti-virus expressing GFP, GFP-tagged wild-type ASXL1$^{1-591}$ and ASXL1$^{1-645}$, or GFP-tagged I581D mutants of ASXL1$^{1-591}$ and ASXL1$^{1-645}$. Whole cell lysates were immunoprecipitated using GFP-trap beads, and immunoprecipitates were analyzed by immunoblotting with antibodies indicated on the left. $n = 2$. **c, d** Log$_2$ fold change heatmap (**c**) and average plots (**d**) showing the BRD4 ChIP-seq signals in ASXL1$^{1-645}$-WT and ASXL1$^{1-645}$-I574D expressing cells compared to GFP control centered on ASXL1$^{1-645}$ peaks. The two clusters were grouped based on the kmeans clustering of the data matrix generated for the heatmap using deepTools (peak number: cluster1, $n = 1730$; cluster2,

$n = 2128$). **e** Track examples showing the chromatin occupancy of endogenous BAP1, endogenous full-length ASXL1, GFP-ASXL1$^{1-645}$-WT, GFP-ASXL1$^{1-645}$-I574D, and BRD4 in ASXL1$^{1-645}$-WT or I574D expressing HEK293T cells on *ECH1* and *POGLUT1* gene promoters. **f** Gene Ontology (GO) enrichment analysis of the 2 clusters as shown in (**c**). Peaks were annotated with TSS regions set to −3, 3 kb using ChIP-seeker. Pathway enrichment of the genes was performed with clusterProfiler with the following settings: fun = "enrichGO", pvalueCutoff = 0.05, pAdjustMethod = "BH". The number next to each cluster represents the total number of genes used for GO analysis in each cluster. Two-sided enrichment tests with Benjamini-Hochberg FDR correction (adj.p) were used. Source data are provided as a Source Data file.

longer variant, ASXL1$^{1-645}$ (Fig. 6d). These results suggested that in the cellular context, the ASXL1$^{1-645}$ cancer-linked truncation variant shows an increased ability to bridge BRD4 to some ASXL1-regulated genes, and the ASXL1$^{1-591}$ truncation may be less efficient in linking BRD4.

**ASXL1$^{1-645}$ but not ASXL1$^{1-591}$ concurrently associates with BRD4 and MLL3/4**

We have previously shown that the amino acid sequence of ASXL1 (aa 607-627) serves as a binding site for the second and third PHD fingers of methyltransferases MLL3 and MLL4, denoted as MBH (MLL binding helix, ASXL1$_{MBH}$)[10,38]. The ASXL1$^{1-645}$ variant contains ASXL1$_{MBH}$ next to

ASXL1$_{EBM}$, however in the shorter variant, ASXL1$^{1-591}$, the ASXL1$_{MBH}$ region is truncated (Fig. 7a). Western blot analysis in HEK293T cells showed that the longer variant, ASXL1$^{1-645}$, but not ASXL1$^{1-591}$ immunoprecipitated endogenous MLL3 and MLL4, whereas both variants immunoprecipitated endogenous BRD4 (Fig. 7b). GFP-ASXL1$^{1-645}$-I574D mutant impaired in binding to BRD4, associated with endogenous MLL3 and MLL4 on par with wt GFP-ASXL1$^{1-645}$, indicating that the interactions of ASXL1 with BRD4 and MLL3/4 might be independent.

To quantitatively assess the relationship of BRD4 and MLL4 binding partners, we measured binding affinities of BRD4$_{ET}$ and MLL4$_{PHD2/3}$ to the region of ASXL1 containing both ASXL1$_{EBM}$ and

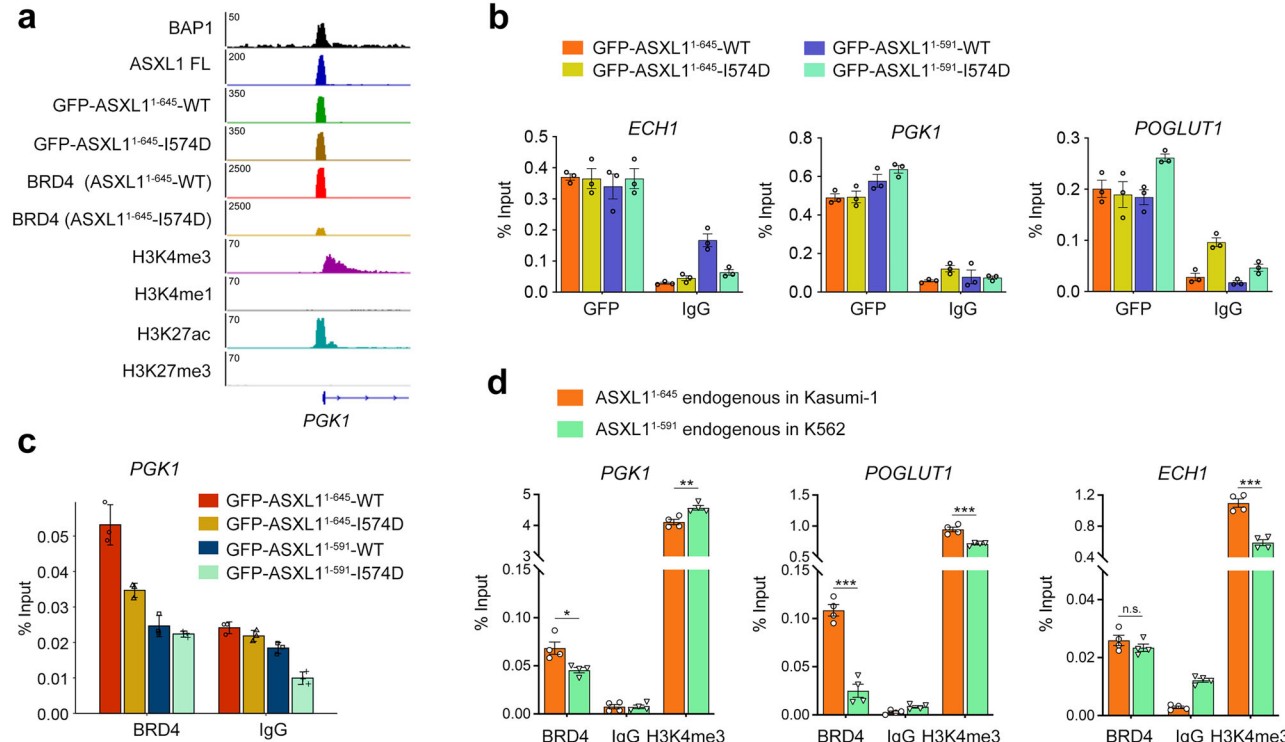

**Fig. 6 | BRD4 binds to promoters of the ASXL1-regulated genes. a** Track examples showing the chromatin occupancy of endogenous BAP1, endogenous full-length ASXL1, GFP-ASXL1[1-645]-WT, GFP-ASXL1[1-645]-I574D, BRD4, and the indicated histone marks in GFP-ASXL1[1-645]-WT or GFP-ASXL1[1-645]-I574D expressing HEK293T cells on *PGK1* promoter. **b** ChIP-qPCR was performed using HEK293T cells transduced with GFP-ASXL1[1-645]-WT or GFP-ASXL1[1-645]-I574D plasmids. Antibody against GFP was used for ChIP, and normal IgG was used as a control. PCR was conducted with primers specific for the promoter regions of the indicated genes. Data represent the mean ± SEM of three independent experiments. **c** ChIP-qPCR was performed as in (**b**) but using an antibody against BRD4 for ChIP. PCR was conducted with primers specific for the indicated gene promoter. Data represent the mean ± SEM of three independent experiments. **d** ChIP-qPCR was performed using Kasumi-1 and K562 cells. Antibodies against BRD4 and H3K4me3 were used for ChIP, and normal IgG was used as a control. PCR was conducted with primers specific for the promoter regions of the indicated genes. Data represent the mean ± SEM of four independent experiments. *$p < 0.05$, **$p < 0.01$, and ***$p < 0.001$ by two-tailed unpaired Student's t test. The $p$-values from left to right are: *PGK1*: 0.016271, 0.007257; *POGLUT1*: $8.99 \times 10^{-5}$, 0.00093; *ECH1*: 0.288788, 0.000235. Source data are provided as a Source Data file.

ASXL1[MBH] (aa 564-635 of ASXL, ASXL1[EBM-MBH]) (Fig. 7c–f). $K_d$s, determined by microscale thermophoresis (MST), revealed that ASXL1[EBM-MBH] was bound by BRD4[ET] as strong as ASXL1[EBM] was bound by BRD4[ET] (Fig. 7d, e). Much like BRD4[ET], MLL4[PHD2/3] recognized ASXL1[EBM-MBH] and ASXL1[MBH] almost equally well (Fig. 7c, e). The binding affinity of MLL4[PHD2/3] toward ASXL1[EBM-MBH] remained unchanged in the presence of BRD4[ET] (Fig. 7e, f), confirming that the interactions of ASXL1 with BRD4 and MLL4 are independent, and the longer variant, ASXL1[1-645] can concurrently associate with both proteins in contrast to the shorter variant ASXL1[1-591] that binds only BRD4.

The AlphaFold modeling of ASXL1[EBM-MBH] suggested that this fragment is disordered except for two regions, amino acids ~568–581 that are in an extended conformation and amino acids 607-627 that adopt an α-helical conformation (Fig. 7g). Interestingly, the former is ASXL1[EBM,] and the latter is ASXL1[MBH], indicating that these regions are performed for binding of BRD4[ET] and MLL4[PHD2/3], as in the respective complexes, ASXL1[EBM] is bound as a β-strand and ASXL2[MBH][10] is bound as an α-helix (Fig. 7g, overlays). Approximately 30 aa separate ASXL1[EBM] and ASXL1[MBH], which should be sufficient to avoid steric hindrance when both BRD4 and MLL4 form complexes with ASXL1.

## Functional correlation of ASXL1 and BRD4

To examine whether ASXL1 and BRD4 correlate in oncogenesis, we performed gene expression analysis using publicly available functional genomic databases. Transcript levels of ASXL1 and BRD4 in six cancer types, including cholangiocarcinoma (CHOL), head and neck squamous cell carcinoma (HNSCC), glioblastoma (GBM), kidney renal clear

cell carcinoma (KIRC), acute myeloid leukemia (AML) and brain lower grade glioblastoma (LGG), revealed a significant upregulation of both ASXL1 and BRD4 in cancer tissues compared to normal tissues (Fig. 8a), and this positive correlation was confirmed through independent analysis of mRNA expression levels from other datasets (Fig. 8b–d and Supplementary Figure 7). Comparison of BRD4 ChIP-seq data from various cancer cell lines and normal cells showed that BRD4 occupies the ASXL1 promoter in all datasets analyzed, with a particularly high level of BRD4 being observed in glioblastoma (Fig. 8e). This analysis suggested that ASXL1 itself could be a transcriptional target of BRD4 and pointed to a possible feed-forward mechanism that would explain the high correlation of ASXL1 and BRD4 in malignancies, particularly in glioblastoma.

GO/Molecular Function (MF) enrichment analysis of the top 1000 genes upregulated along with BRD4 or ASXL1 in the LGG and KIRC datasets indicated a high degree of association with DNA binding, chromatin organization and transcriptional regulation (Fig. 8f, g and Supplementary Fig. 8). Furthermore, BRD4 and ASXL1 seem to share similar functional profiles and may have coordinated regulatory roles, as their top 50 enriched GO/MF terms showed strong positive correlation (Fig. 8h).

GO/MF terms with high representation ratios (which is defined as the number of upregulated genes for a given GO/MF relative to the total number of genes of that GO/MF) in combination with false discovery rate (FDR) and fold enrichment metrics have been used to distinguish strong functional relevance. Using this approach, we identified histone binding, chromatin binding, and transcription

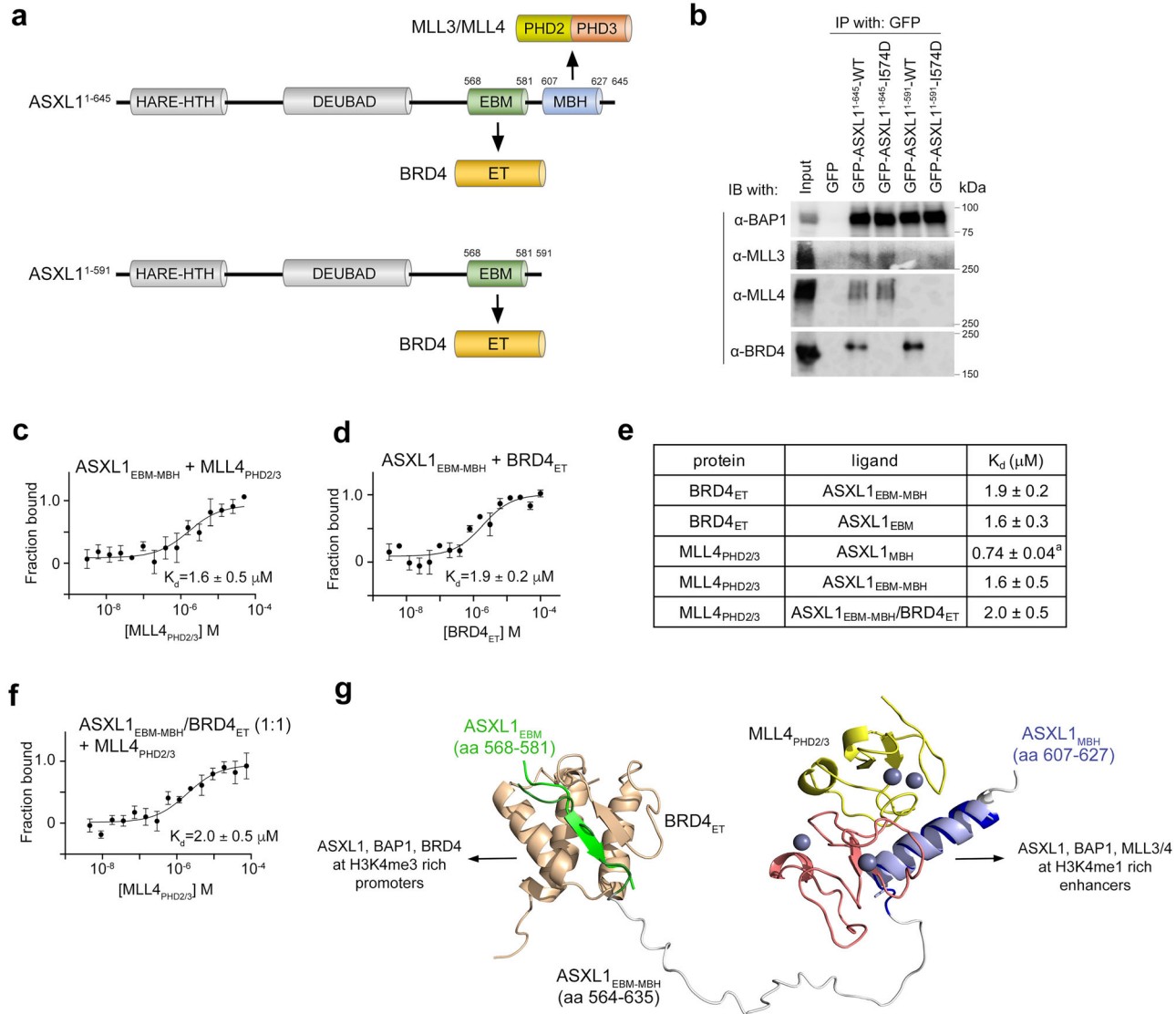

**Fig. 7 | ASXL1^1-645 but not ASXL1^1-591 concurrently and independently associates with BRD4 and MLL3/4. a** Schematics of the cancer-relevant variants ASXL1^1-591 and ASXL1^1-645 and their binding partners, BRD4 and MLL3/4. MBH (MLL binding helix, ASXL1_MBH). **b** HEK293T cells were transduced with lenti-virus expressing GFP, GFP-tagged wild-type ASXL1^1-591 and ASXL1^1-645, or GFP-tagged I581D mutants of ASXL1^1-591 and ASXL1^1-645. Whole cell lysates were immunoprecipitated using GFP-trap beads, and immunoprecipitates were analyzed by immunoblotting with antibodies indicated on the left. $n = 2$. **c, d** MST binding curves used to determine binding affinities of MLL4_PHD2/3 (**c**) and BRD4_ET (**d**) for ASXL1_EBM-MBH. The K_d values represent average of three independent measurements, and errors represent SEM. $n = 3$ **e** Binding

affinities of BRD4_ET and MLL4_PHD2/3 for the indicated ASXL1 constructs measured by MST. (^a) taken from (^10). **f** An MST binding curve used to determine binding affinity of MLL4_PHD2/3 for ASXL1_EBM-MBH in the presence of BRD4_ET (1:1 molar ratio). The K_d value represents average of three independent measurements, and error represents SEM. $n = 3$ (**g**). Overlay of the AlphaFold predicted model of ASXL1_EBM-MBH taken from UniProt #Q8IXJ9 (ASXL1_EBM is dark green, ASXL1_MBH is dark blue, and other fragments are grey) with the structure of BRD4_ET (wheat) in complex with ASXL1_EBM (light green) (this work) and the structure of MLL4_PHD2/3 (yellow and salmon) in complex with ASXL2_MBH (light blue) (PDB 9ATN). Source data are provided as a Source Data file.

coregulator activity as significant functions for both BRD4 and ASXL1 (Fig. 8i, j). Collectively, these findings reinforce the notion that ASXL1 and BRD4 share their roles in chromatin-associated regulatory processes that may be dysregulated in cancer.

In conclusion, in this study, we report the molecular mechanism by which BRD4 is recruited to promoters of ASXL1 gene targets. We show that BRD4_ET binds to the ASXL1_EBM motif following the DEUBAD domain of ASXL1 and that this interaction is necessary for BRD4 to co-localize with ASXL1 at active promoters. We also found that BRD4_ET has a DNA-binding activity and that the ASXL1-binding and DNA-binding sites do not overlap, indicating two independent functions of BRD4_ET. We demonstrate that the cancer-related truncated variants ASXL1^1-645 and ASXL1^1-591 retain BRD4 binding function, with ASXL1^1-645 showing an enhanced ability to recruit BRD4 to ASXL1-occupied promoter regions.

In contrast to ASXL1^1-591, ASXL1^1-645 concomitantly associates with BRD4_ET and MLL4_PHD2/3. In addition to the BRD4-binding site, ASXL1^1-645 contains the MLL3/4-binding site, which is missing in the shorter ASXL1^1-591 variant. The ability of ASXL1^1-645 to engage with BRD4 and MLL3/4 may help in the recruitment to or stabilization of the BAP1 complex at specific genomic regions, i.e., at H3K4me3-rich active promoters through binding of BRD4, and at H3K4me1-rich active enhancers through binding of MLL4[10,38] (Fig. 7g).

Our mechanistic findings can provide a rational approach for developing cancer treatments through targeting the ASXL1-BRD4 interaction with small molecule inhibitors, because the cancer-specific truncation variants of ASXL1 but not full-length ASXL1 have previously been shown to appreciably interact with BRD4 and induce myeloid malignancies in animal models[6,26,39]. Although it remains unknown as

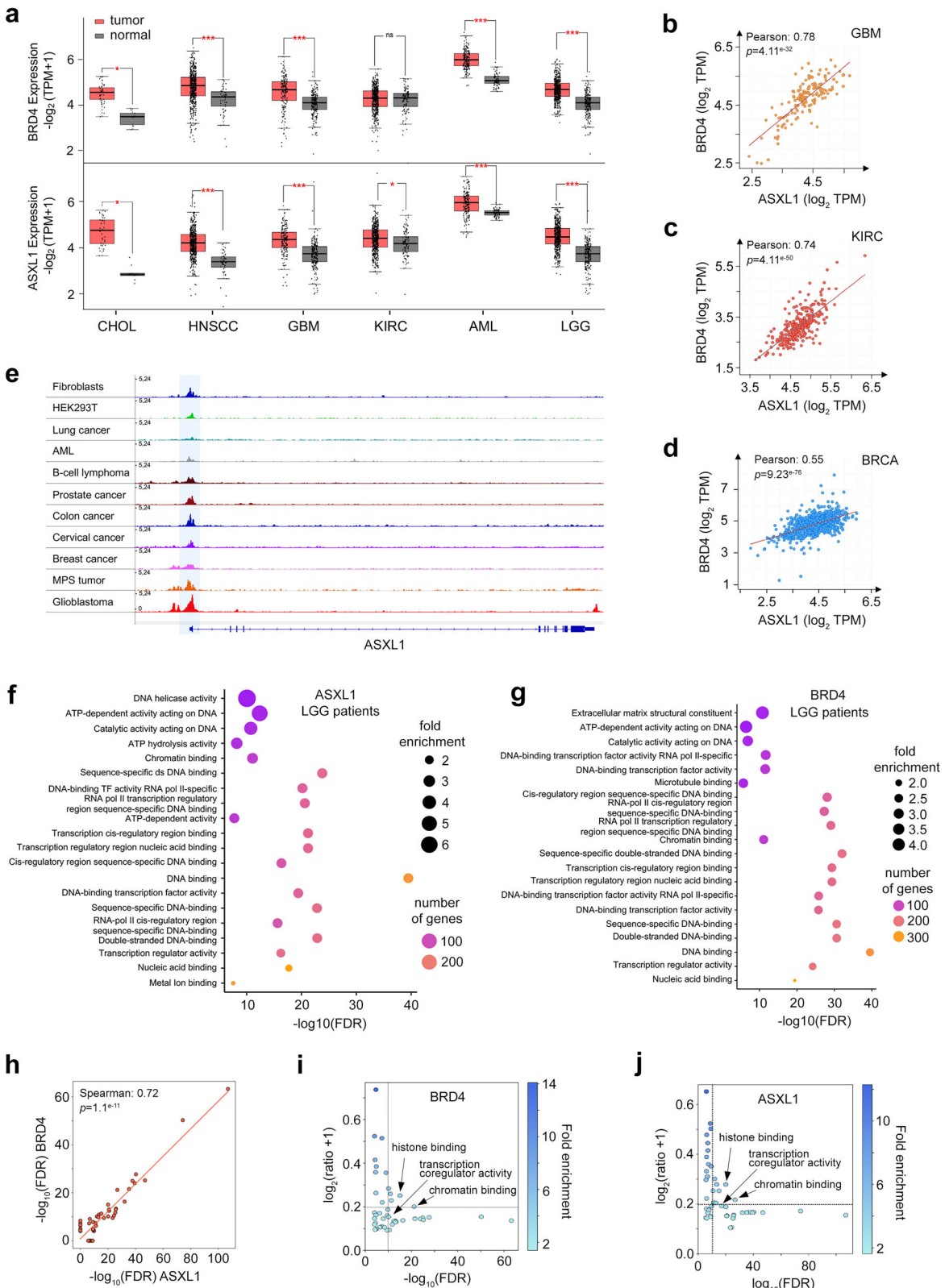

to how the inhibition of the interaction with BRD4 occurs in the context of full-length ASXL1, the intramolecular contacts with the C-terminal region of ASXL1 that occlude $\text{ASXL1}_{EBM}$ could play a role. Another possibility is the regulation of the BRD4 binding and chromatin recruitment by other proteins that interact with ASXL1. An exceedingly long disordered region encompassing almost 1,000 residues of ASXL1 between the N-terminal folded domains and the C-terminal zinc finger could harbor binding sites and motifs for other yet to be discovered binding partners of ASXL1. Functions of other than $\text{BRD4}_{ET}$ and $\text{MLL4}_{PHD2/3}$ domains in BRD4 and MLL3/4, such as acetyllysine binding activity of bromodomains of BRD4[18–20] and TET3- and H4K16ac-binding activities of MLL3/4[40–43] may also contribute.

Our genomic analysis from six cancer types reveals a strong positive ASXL1-BRD4 correlation and a high level of BRD4 itself on the

**Fig. 8 | EBM of ASXL1 is required for the recruitment of BRD4 to the ASXL1-bound gene promoters. a** Expression levels of ASXL1 and BRD4 across various cancer types (red) compared to normal tissues (grey) based on publicly available transcriptomic datasets available at GEPIA. The plot illustrates the relative expression levels of ASXL1 and BRD4, highlighting the same pattern of upregulation in different cancer types versus normal tissues. Each data point represents an independent biological sample (patient-derived tumor or normal tissue). Box plots display the median as the center line, the 25th and 75th percentiles as the lower and upper bounds of the box, and whiskers extending to the most extreme values within 1.5 × the interquartile range. Individual data points are overlaid using jitter to show the distribution of all samples. Cholangiocarcinoma (CHOL), Head and Neck Squamous Cell Carcinoma (HNSCC), Glioblastoma (GBM), Kidney Renal Clear Cell Carcinoma (KIRC), Acute Myeloid Leukemia (AML), Brain Lower Grade Glioblastoma (LGG). Two-sided Student's t-test p-value: ns = $p > 0.5$; *$p < 0.05$, **$p < 0.01$, ***$p < 0.001$. **b–d** Correlation analysis of BRD4 and ASXL1 mRNA expression across various cancer types. Pearson correlation coefficients and exact two-sided p-values are indicated. GBM Glioblastoma, KIRC Kidney Renal Clear Cell, BRCA Breast cancer. **e** BRD4 ChIP-seq analysis across various cell lines. The plot represents BRD4 binding peaks on the ASXL1 gene, showing differential enrichment patterns in different cancer contexts compared to normal cells. **f, g** Gene Ontology (GO) enrichment analysis of the top 1,000 genes upregulated along with ASXL1 or BRD4, in LGG patients. GO terms are ranked by statistical significance ($-\log_{10}$ FDR). Dot size reflects the fold enrichment, while color indicates the number of genes annotated to each term. **h** Scatter plot showing correlation of the top 50 enriched GO/MF terms between ASXL1 and BRD4 in LGG. Spearman correlation coefficient, exact two-sided p value, and regression line are shown. **i, j** GO/MF terms for ASXL1 and BRD4 were plotted by $-\log_{10}$(FDR), fold enrichment and $\log_2$((ratio nGenes/Total genes of GO/MF term) + 1). The dashed lines divide the most significant GO/MF: $-\log_{10}$(FDR) > 10 and $\log_2$ representation ratio > 0.20 (15% representation).

ASXL1 promoter region. The latter data suggest a possible feed-forward mechanism, because BRD4 is a well-recognized master regulator of transcription[23–25]. It will be essential in future studies to explore these mechanisms and to determine the role of other binding partners of BRD4 and ASXL1 in the formation of the BRD4-ASXL1 complex. Future work is also needed to better understand how the BRD4/ASXL1/MLL3/4 epigenetic machinery and its associated transcriptional programs can be therapeutically targeted for cancer treatment.

## Methods

### Protein expression and purification
BRD4$_{ET}$ domain (aa 601-683), ASXL1$_{EBM-MBH}$-6×His (aa 635-564) and MLL4$_{PHD2/3}$ (aa 227–324) in a pGEX-6P-1 vector was used in this study. The ET domain mutants of BRD4 (E651K, E653K, E651K/E653K, R665E, R669E, and R676A/K677A/K678A/R679A/K680A) were generated using Agilent QuikChange Lightning Site-Directed Mutagenesis kit. A tryptophan was introduced to the C-terminus of BRD4$_{ET}$ for fluorescence spectroscopy studies. All constructs were confirmed by DNA sequencing. Unlabeled and $^{15}$N-labeled proteins were expressed in *E. coli* BL21 (DE3) RIL (Agilent) cells grown in Luria Broth or minimal media supplemented with $^{15}$NH$_4$Cl (Sigma-Aldrich). Protein production was induced with 0.5 mM isopropyl β-D-1-thiogalactopyranoside (IPTG) at 16 °C for 18 h. Bacteria were harvested by centrifugation and lysed by sonication in a buffer consisting of 50 mM Tris pH 7.5, 500 mM NaCl, 5 mM DTT, PMSF and DNase I. GST-fusion proteins were purified on glutathione agarose 4B beads (Thermo Fisher Sci). The GST-tag was cleaved on-resin with human rhinovirus (HRV) 3 C protease at 4 °C overnight. Proteins were further purified by size exclusion chromatography using HiPrep 16/600 Superdex 75 columns (GE Healthcare) in buffer 50 mM Tris pH 7.5, 150 mM NaCl, and 5 mM DTT. The proteins were concentrated in Millipore concentrators and stored at −80 °C. MLL4$_{PHD23}$ (aa 227-324) was produced as described in ref. 10.

### ASXLs peptides
The following peptides were synthesized by Synpeptide and used in NMR, fluorescence spectroscopy, MST and ITC experiments: Ac-KVPPIRIQLSRIKP-NH$_2$ (aa 568-581 of ASXL1), Ac-KVPPLKIPVSRISP-NH$_2$ (aa 615-628 of ASXL2), Ac-RVPPLKIQLSKIGP-NH$_2$ (aa 1008-1021 of ASXL3), Ac-KVPPIDIQLSRIKP-NH$_2$ (R573D), Ac-KVPPIRDQLSRIKP-NH$_2$ (I574D).

### NMR titration experiments
NMR experiments were performed at 298 K on a Varian INOVA 600 MHz spectrometer equipped with a cryoprobe. $^1$H,$^{15}$N HSQC spectra of 100 μM $^{15}$N-labeled BRD4$_{ET}$ in 25 mM Tris pH 7.0, 150 mM NaCl, 1 mM TCEP, 10% D$_2$O were recorded in the presence of increasing concentrations of ASXLs peptides. NMR data were processed and analyzed with NMRPipe and NMRDraw as described in ref. 44. The K$_d$

values were calculated using nonlinear least-squares analysis and the equation:

$$\Delta\delta = \Delta\delta_{max}\left(([L]+[P]+K_d) - \sqrt{([L]+[P]+K_d)^2 - 4[P][L]}\right)/2[P] \quad (1)$$

where $[L]$ is the concentration of the peptide, $[P]$ is concentration of the protein, $\Delta\delta$ is the observed chemical shift change, and $\Delta\delta_{max}$ is the normalized chemical shift change at saturation. Normalized chemical shift changes were calculated using the equation:

$$\Delta\delta = \sqrt{(\Delta\delta H)^2 + (0.154 \cdot \Delta\delta N)^2} \quad (2)$$

where $\Delta\delta$ is the change in chemical shift in parts per million (ppm).

### Isothermal titration calorimetry
Experiments were carried out on a MicroCal auto-ITC200 instrument at 25 °C while stirring at 750 rpm in the ITC buffer of pH 6.5, consisting of 100 mM sodium phosphate, 2 mM EDTA and 2 mM β-mercaptoethanol. Peptide concentration was determined by weight and confirmed by NMR, and protein concentrations by A280 measurements. BRD4$_{ET}$ sample (0.2 mM) was placed in the cell, whereas the micro-syringe was loaded with the ASXL1 peptide (2.0 mM) in the ITC buffer. The titrations were conducted using 17 successive injections of 2.4 μL (the first at 0.4 μL and the remaining 16 at 2.4 μL) with a duration of 4 s per injection and a spacing of 180 s between injections. The collected data was processed using the Origin 7.0 software program (OriginLab) supplied with the instrument according to the "one set of sites" fitting model.

### Fluorescence spectroscopy
Spectra were recorded at 298 K on a Fluoromax-3 spectrofluorometer (HORIBA), as described in ref. 40 with the following modifications. The samples of tryptophan-containing 1 μM BRD4$_{ET}$ in 25−50 mM Tris pH 7.0−7.5, 150 mM NaCl, and 0−1 mM TCEP (with or without 75 nM of 147 bp 601 DNA) and progressively increasing concentrations of peptides were excited at 295 nm. Emission spectra were recorded between 380 and 440 nm with a 0.5 nm step size and a 0.3−1 s integration time. The K$_d$ values were determined using nonlinear least-squares analysis and the equation:

$$\Delta I = \Delta I_{max}\left(([L]+[P]+K_d) - \sqrt{([L]+[P]+K_d)^2 - 4[P][L]}\right)/2[P] \quad (3)$$

where $[L]$ is concentration of the peptide, $[P]$ is concentration of the protein, $\Delta I$ is the observed change of signal intensity and $\Delta I_{max}$ is the difference in signal intensity of the free and bound states of the protein. K$_d$ values is the average of at least three separate experiments and the error was calculated as the standard deviation between the runs.

## Microscale thermophoresis (MST)

MST experiments were carried out on a Monolith NT.115 instrument (NanoTemper). All experiments were performed using SEC-purified ASXL1$_{EBM-MBH}$-6×His (aa 635–564) in 25 mM MES pH 6.8 buffer, 150 mM NaCl, and 3.0 mM DTT. This protein was labeled using a His-Tag Labeling Kit RED-tris-NTA (2nd Generation, NanoTemper) and kept constant at 10 nM. Dissociation constants were determined using a direct binding assay in which MLL4$_{PHD23}$ and BRD4$_{ET}$ proteins were varied in concentration by serial dilution of discrete samples. The measurements were performed at 50 % LED and medium MST power with 3 s pre-laser time, 20 s laser on-time, and 1 s off-time. The $K_d$ values were calculated using MO Affinity Analysis software (NanoTemper) using a 1:1 stoichiometry and averaged over three separate experiments with error reported as SEM. Plots were generated in GraphPad PRISM.

## NMR structure of the BRD4$_{ET}$-ASXL1$_{EBM}$ complex

NMR samples of 0.5 mM BRD4$_{ET}$ and 1.5 mM ASXL1$_{EBM}$ peptide (aa 568-581, Suppl. Table 1) were prepared in 100 mM sodium phosphate buffer (pH 6.5) supplemented with 5 mM perdeuterated DTT and 0.5 mM EDTA in H$_2$O/$^2$H$_2$O (9/1) or $^2$H$_2$O. All NMR spectra were acquired at 25 °C on Bruker 600 MHz and 800 MHz spectrometers equipped with Z-gradient triple-resonance cryoprobes (Bruker TopSpin v3.0). The backbone $^1$H, $^{13}$C, and $^{15}$N resonances were assigned using three-dimensional triple-resonance HNCA, HN(CO)CA, HN(CA)CB, and HN(COCA)CB experiments. The side-chain atoms were assigned from three-dimensional HCCH-TOCSY, HCCH-COSY, and (H)C(CO)NH-TOCSY. The NOE-derived distance restraints were obtained from three-dimensional $^{15}$N- or $^{13}$C-edited NOESY spectra. The ASXL1$_{EBM}$ peptide was assigned from two-dimensional TOCSY, NOESY, ROESY, and $^{13}$C/$^{15}$N-filtered TOCSY and NOESY. The intermolecular NOEs were obtained from three-dimensional $^{13}$C-edited (F1), $^{13}$C/$^{15}$N-filtered (F3) NOESY spectra of $^{13}$C/$^{15}$N-labeled BRD4$_{ET}$ in complex with unlabeled ASXL1$_{EBM}$. Spectra were processed with NMRPipe (v2.0)[45] and analyzed using NMRVIEW (v5.0)[46].

Structures of the BRD4 ET domain/ASXL1 peptide (aa 568-581) were calculated with a distance-geometry simulated annealing protocol with CNS. Initial protein structure calculations were performed with manually assigned NOE-derived distance constraints. Hydrogen-bond distance, φ and ψ dihedral-angle restraints from the TALOS+ (v3.70F1) prediction were added at a later stage of structure calculations for residues with characteristic NOE patterns. The converged structures were used for the iterative automated NOE assignment by ARIA refinement[47]. Structure quality was assessed with CNS (v1.5), ARIA (v2.0), and PROCHECK (v.3.5.4) analysis[46–48]. A family of 200 structures was generated, and 20 structures with the lowest energies were selected for the final analysis.

## EMSA

EMSAs were performed by mixing increasing amounts of wt or mutated BRD4$_{ET}$ (with or without ASXL1 peptide at a 1:1 molar ratio) with 10 nM/lane of 147 bp 601 DNA in 20–25 mM Tris pH 7.5, 150 mM NaCl and 5 mM DTT in a 10 μL reaction volume. The reaction mixtures were incubated at room temperature for 30 min (2 μL of loading dye was added to each sample) and loaded onto a 8% native polyacrylamide gel. Electrophoresis was performed in 0.5x TBE buffer at 80–100 V on ice. The gels were stained with SYBR Gold (Thermo Fisher) and visualized by UltraThin LED Illuminator. Uncropped gels are shown in the Source Data file.

## RNA binding assays

Full-length FLAG-BRD4 ΔBDs was expressed in Sf9 insect cells and purified using FLAG M2 agarose (Sigma). FLAG-BRD4 (aa 1–722) was expressed in *E. coli* and purified using FLAG resin (Sigma), as described in ref. 30. Primers were designed to amplify the *PNP* mRNA and *MYC*

eRNA. The RNA probe sequences, and the primer sequences used for PCR amplification are listed in the Source Data file. The T7 promoter sequence was included into the forward primer, and genomic fragments were PCR amplified from SW480 cDNA, sequence-verified, and used as templates for in vitro transcription with the T7 RiboMAX Express Large-Scale RNA Production System (Promega) per the manufacturer's instructions. Synthesized RNAs were purified using Micro-Spin G-25 Columns (GE Healthcare Life Sciences), quantified by Nanodrop (Invitrogen), and verified on a 5% TBE urea gel stained with SYBR Gold (Life Technologies) for 20 min prior to imaging using the ChemiDoc Imaging system (Bio-Rad Laboratories). RNA probes were refolded by heating at 95 °C for 5 min followed by snap-cooling on ice for 5 min. Cold RNA refolding buffer (10 mM Tris-HCl at pH 7, 100 mM KCl, 10 mM MgCl$_2$) was then added, and samples were incubated at room temperature (20–25 °C) for 20–30 min. to allow refolding.

FLAG-tagged BRD4 proteins were incubated with 500 ng of refolded RNA while rotating at 4 °C for 1 h in RNA binding buffer (20 mM Tris-HCl at pH 7.4, 100 mM KCl, 0.2 mM EDTA, 0.05% NP40, 0.4 U RNase inhibitor, PICs). Protein-RNA complexes were recovered using FLAG M2 agarose beads for 1 h at 4 °C. Beads were washed three times with RNA wash buffer (20 mM Tris-HCl at pH 7.4, 200 mM KCl, 0.2 mM EDTA, 0.05% NP40, 0.4 U RNaseOUT, PICs). RNA samples were eluted using Trizol reagent (Invitrogen), resolved on a denaturing 10% TBE-urea gel, stained with SYBR gold for 20 min, and imaged using a ChemiDoc Imaging system (Bio-Rad Laboratories).

## Immunoprecipitation–mass spectrometry

K562 cells (ATCC) stably expressing Flag-Strep ASXL1 were maintained in large-scale culture with RPMI-1640 supplemented with 5% NBS and 1% penicillin/streptomycin. Purification from nuclear extracts was performed according to Nakatani's protocol with some modifications[49]. Briefly, cells were grown until a density of 5–8 x 10$^5$/ml is reached, harvested by centrifugation, then nuclear extracts were prepared and stored at −80 °C. To purify Flag-Strep ASXL1, nuclear extracts were incubated overnight at 4 °C on rotator with anti-Flag M2 agarose (Sigma-Aldrich) in wash buffer, 50 mM Tris–HCl, pH 7.3, 150 mM KCl, 5 mM MgCl$_2$, 0.2 mM EDTA, 10% glycerol, 0.1% Tween, 2 mM β-mercaptoethanol, 0.25 mM PMSF and proteases inhibitors (Sigma). Bound proteins were washed several times, then eluted from beads with Flag peptide at 200 μg/ml in wash buffer (Sigma) and incubated with Strep-Tactin beads in wash buffer at 4 °C overnight. Bound proteins were similarly washed several times before elution with D-Biotin (Sigma). Purified fractions were run on 4–12% Bis-Tris NuPAGE gels (Invitrogen) and bands were cut for downstream tryptic digestion of proteins and MS/MS protein identification performed at the Taplin Biological Mass Spectrometry Facility (Boston, MA).

## ChIP-seq

For ChIP-seq assays, parental and stably expressing GFP, GFP-ASXL1$^{1-645}$-WT or GFP-ASXL1$^{1-645}$-I574D HEK293T cells (ATCC) were maintained in Dulbecco's Modified Eagle Medium (Gibco), supplemented with 10% FBS (Sigma), 1% penicillin-streptomycin, and 2 mg/ml puromycin. Cells were cultured in tissue culture-treated flasks at 37 °C, 5% CO$_2$. The cells were split at ~70–90% confluency using 0.05% trypsin-EDTA for cell detachment. ChIP-seq was performed as described in ref. 27. The cell pellets were collected and washed twice with ice-cold PBS and fixed with 1% paraformaldehyde for ten minutes at room temperature. The paraformaldehyde solution was quenched with 2.5 M (1/20) glycine, and the cell pellets were washed twice with PBS. The cell pellets were resuspended in lysis buffer (50 mM HEPES, pH = 7.5, 140 mM NaCl, 1 mM EDTA, 10% Glycerol, 0.5% NP-0.4, 0.25% Triton X-100 with protease inhibitors), incubated on nutator for 10 min at 4 °C and centrifuged at 500 g for 5 min. Supernatant was discarded, and the cell pellets were washed with wash buffer (10 mM Tris-HCl, pH = 8.0, 200 mM NaCl, 1 mM EDTA, 0.5 mM EGTA with protease

inhibitors) and resuspended in the sonication buffer (10 mM Tris-HCl, pH = 8.0, 1 mM EDTA, 0.1% SDS with protease inhibitors). Sonication was performed with Covaris tubes which were set to 7.5% duty factor, 175 peak intensity power, and 200 cycles per burst for 120–600 s. The 10×dilution buffer (10% Triton x-100, 1 M NaCl, 1% Na-Deoxycholate, 5% N-Lauroylsarcosine, 5 mM EGTA) was added to the lysate, samples were centrifuged at maximum speed at 4 °C for 20 min, and the following antibodies were added: α-GFP (Santa Cruz, sc-9996, 5 µg per sample), α-BRD4 (Fortis, A700-004, 5 µg per sample), α-ASXL1 (made inhouse, 40 µL anti-sera per sample), or α-BAP1 (made inhouse, 40 µL anti-sera per sample). After incubation at 4 °C overnight, 100 µl protein A/G agarose beads (Santa Cruz, sc-2003) were added for 4 h. The beads were washed 4 times with ice-cold RIPA buffer (50 mM HEPES, pH = 7.5, 500 mM LiCl, 1 mM EDTA, 1.0% NP-40, 0.7% Na-Deoxycholate), followed by once with ice-cold TE buffer (with 50 mM NaCl). DNA for each IP sample was eluted with the elution buffer (50 mM Tris-HCl, pH = 8.0, 10 mM EDTA, 1.0% SDS) and reverse cross-linked at 65 °C oven for 13.5 h, followed by protease K digestion at 55 °C for 2 h. The genomic DNA fragments were then further purified with Qiagen DNA purification kit (Qiagen, 28104). Libraries were prepared by using the KAPA HTP library preparation kit (Roche, 07961901001) and sequenced on the Illumina NovaSeq 6000.

## ChIP-seq analysis

For ChIP-seq data analysis shown in Fig. 5, all the peaks were called with the MACS v2.1.0 software using default parameters and corresponding input samples. Metaplots and heatmaps were generated by utilizing ngsplot database to display ChIP-seq signals. Peak annotation, motif analysis, and super enhancer analysis were performed with HOMER and ChIPseeker. Pathway analysis was performed with Metascape.

For ChIP-seq analysis shown in Figs. 1 and 6, samples were mapped to hg38 using Bowtie2 and peaks were called with MACS v2.0 relative to GFP-transfected cells as a control. Peak genomic distribution was analyzed using the ChIPseeker R package, and heatmaps were generated with deepTools v.3.5.6. ChIP-seq tracks were visualized with Integrative Genomics Viewer (IGV) v.2.16.2.

## ChIP-qPCR

For ChIP-qPCR assays, Kasumi-1 cells (ATCC) were cultured in RPMI-1640 medium (Gibco) supplemented with 20% fetal bovine serum (FBS; Gibco), 1% penicillin-streptomycin (Gibco), and incubated at 37 °C in a humidified atmosphere of 5% $CO_2$. Cells were cultured in suspension and subcultured every 2–3 days to maintain the density at $0.2–1 \times 10^6$ cells/ml.

K562 cells were cultured in RPMI-1640 medium (Gibco) with 10% FBS and 1% penicillin-streptomycin. Cells were cultured in suspension at 37 °C with 5% $CO_2$ and were passaged every 2–3 days to be in exponential growth, keeping cell density at $0.2–1 \times 10^6$ cells/ml.

HEK293T cells were grown in Dulbecco's Modified Eagle Medium (DMEM; Gibco) supplemented with 10% FBS and 1% penicillin-streptomycin. Cells were adherently cultured in tissue culture-treated flasks at 37 °C, 5% $CO_2$. Medium was replaced every 2–3 days, and cells were split at ~70–90% confluency using 0.05% trypsin-EDTA for cell detachment.

ChIP assays were performed using Kasumi-1 and K562 cells, as well as HEK293T cells transduced with GFP-ASXL1[1-645]-WT, GFP-ASXL1[1-645]-I574D, GFP-ASXL1[1-591]-WT or GFP-ASXL1[1-591]-I574D ($5 \times 10^6$ cells per condition), following protocols[39]. Genomic DNA regions of interest were immunoprecipitated using antibodies against BRD4 (Bethyl Laboratories, A301-985A50), H3K4me3 (Diagenode, C15410003), or GFP (Takara Bio, USA632592). Immune complexes were washed and eluted from the beads using SDS buffer, followed by treatment with RNase and proteinase K. Crosslinks were reversed by overnight incubation at 65 °C, and the ChIP DNA was purified using phenol-chloroform extraction and ethanol precipitation. DNA concentration

was quantified using a Qubit 3.0 fluorometer, and the purified DNA was subsequently used for qPCR. The primers used for ChIP-qPCR were:

hPGK1-F1: GAATCACCGACCTCTCTCCC; hPGK1-R1: TACCTCA-TAACGACCCGCTT; hPOGLUT1-F1: GGCAACTGGTCCATTCGTTT; hPOGLUT1-R1: CACAAAGATGGCCACCGTG; hECH1-F1: CGGGGA-TAGTGGCTTCTCG; hECH1-R1: TCCCCGTCCTACATCTGCTA; hVIM-F1: AAAACTTAGGGGCGCTCTTG; hVIM-R1: ATTCAAGTCTCAGCGGGCT.

## Immunoprecipitation and Western blot

HEK293T cells were collected and lysed in lysis buffer (50 mM Tris, pH 8.0, 150 mM NaCl, 0.5% Triton X-100, 10% glycerol, protease inhibitors and benzonase) and incubated on ice for 30 min. The lysate was then centrifuged at 20,000 g at 4 °C for 15 min, the supernatant was collected to remove any cell debris. The protein concentration was measured using Nanodrop (protein A280) and samples were normalized and the final concentrations of each sample and sample buffer were calculated and 5x loading dye was added. The cell lysate was incubated with GFP-trap beads (Bulldog Bio) at 4 °C overnight with rotation. The beads were then washed four times with ice-cold lysis buffer, and the bound proteins were eluted with 0.1 M glycine-HCl, pH 2.5, boiled in 5× SDS sample loading buffer and subjected to SDS-PAGE electrophoresis. The proteins were transferred from the gel to a nitrocellulose membrane at 350 mA at 4 °C for 90 min. The membrane was blocked with 5% PBST-containing non-fat dry milk (Fisher Scientific) for 30 min to remove any non-specific binding to the antibody. The membrane was then incubated with the primary antibodies for 2 hours at room temperature. After three washes with PBST, the membrane was incubated with the secondary antibody mouse anti-rabbit IgG, light chain specific (Jackson Immunoresearch, 211-032-171) at a 1:10,000 dilution for 1 hour at room temperature. Finally, the membrane was washed again 3 times at 10-minute intervals, and a HRP substrate (Millipore Sigma) was added to the membrane and incubated for 60 s. The membrane was exposed to a digital imager for chemiluminescent detection. Antibodies BAP1, (Cell Signaling, 13271), BRD4 (Cell Signaling, 13440), GAPDH (Santa Cruz, sc-32233) and GFP (Santa Cruz, sc-9996) were used at a 1:2,000 dilution. Anti-MLL3 and anti-MLL4 polyclonal antibodies were gift from A. Shilatifard.

## Public cancer data acquisition and analysis

Tumor vs normal expression profiling: BRD4 and ASXL1 expression (TPM) across tumors and normal tissues were analyzed using the GEPIA web server (http://gepia2.cancer-pku.cn/). Datasets are available from The Cancer Genome Atlas program and The Genotype-Tissue Expression (GTEx) project. Tumors are annotated according to lineage. Normal tissues are typically normal adjacent tissues (NAT) with associated blood samples. Six cancer types were analyzed: LGG: Brain Lower Grade Glioblastoma, CHOL: Cholangiocarcinoma, HNSCC: Head and Neck Squamous Cell Carcinoma, GBM: Glioblastoma, KIRC: Kidney Renal Clear Cell Carcinoma, and AML: Acute Myeloid Leukemia. Numbers of samples for tumor and normal tissues for each lineage are: LGG (tumor $n = 518$, normal $n = 207$), CHOL (tumor $n = 36$, normal $n = 9$), HNSCC (tumor $n = 519$, normal $n = 44$), GBM (tumor $n = 163$, normal $n = 207$), KIRC (tumor $n = 523$, normal $n = 100$), and AML (tumor $n = 173$, normal $n = 70$), as obtained by combining TCGA and GTEx data.

Correlation analysis: Correlation analysis of six datasets from TCGA and CPTAC databases (LGG, KIRC, HNSCC, CHOL, BRCA, and GBM) was performed using the cBioPortal cancer genomics platform (http://cbioportal.org). Pearson rank correlations between BRD4 and ASXL1 mRNA levels across these tumor types were calculated. Reported values are the correlation coefficient and *p*-value.

ChIP-seq data analysis: BRD4 ChIP-seq datasets were retrieved from Cistrome Data Browser (http://cistrome.org/db). GSM2359439 (Lung cancer, A549 cells), GSM2330607 (Breast cancer, SUM229 cells),

GSM1890758 (Colon cancer, COLO-741 cells), GSM2495660 (Prostate, 22Rv1 cells), GSM1249906 (Cervical cancer, Hela cells), GSM1527929 (MPS tumor, 90-8TL cells), GSM1195551 (Glioblastoma, U87 cells), GSM1296634 (B-cell lymphoma, OCI-Ly1 cells), GSM2716697 (AML, MV4-11 cells), GSM1249882 (HEK293T cells), GSM1915116 (Fibroblasts, IMR90 cells) were aligned to hg38. Coverage over the ASXL1 locus (chr20:31,900,000–32,000,000) was extracted and plotted as profile tracks with IGV_2.19.4.

Gene Ontology enrichment and comparative analysis: LGG and KIRC datasets were analyzed with cbioportal (http://cbioportal.org). Patients were grouped into quartiles of BRD4 or ASXL1 mRNA expression to retrieve the top 1000 genes upregulated along with ASXL1 or BRD4. These genes were subjected to GO enrichment/ Molecular Function using the ShinyGO 0.82 platform. Terms with FDR ≤ 0.05 were retained. The top 50 enriched GO/MF for BRD4 and ASXL1 were correlated by their -log10 (FDR). Pearson or Spearman coefficient of correlation and $p$-value were calculated. Comparative analysis was performed to identify biologically significant GO/MF terms. Fold enrichment, log2((ratio of observed genes over total genes of that GO/MF term) +1), and $-\log_{10}$(FDR) were extracted and plotted for BRD4 and ASXL1.

### Statistics and reproducibility
Tryptophan fluorescence, ITC, MST data are presented as average of at least three independent measurements ± SD (± SEM in MST). RNA binding experiments were performed in three biological replicates and showed identical results. EMSA experiments were performed at least twice.

### Reporting summary
Further information on research design is available in the Nature Portfolio Reporting Summary linked to this article.

## Data availability
Coordinates for the BRD4$_{ET}$-ASXL1$_{EBM}$ complex have been deposited in the Protein Data Bank under accession number 9VQ1. NMR parameters for the BRD4$_{ET}$-ASXL1$_{EBM}$ complex have been deposited to the Biological Magnetic Resonance Bank under accession number 36770. The genomics data for endogenous full-length ASXL1, BRD4 and BAP1, and for GFP-ASXL1$^{1-645}$-WT, GFP-ASXL1$^{1-645}$-I574D, BRD4 in GFP-ASXL1$^{1-645}$-WT and BRD4 in GFP-ASXL1$^{1-645}$-I574D generated in this study have been deposited in the Gene Expression Omnibus database under accession code GSE302969. Another dataset previously generated by us and used in this study (histone H3 PTMs in HEK293T cells) is available under accession code GSE196860. The mass spec data have been deposited to the PRIDE database [http://www.ebi.ac.uk/pride] under accession number PXD073614. All other relevant data supporting the key findings of this study are available within the article and its Supplementary Information files. Source data are provided with this paper.

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

## Acknowledgements

We thank Salima Daou and Haithem Barbour for technical help with the ASXL1 purification and initial analysis. This work was supported in part by grants from NIH AG067664, CA252707 and GM157928 to T.G.K., GM146979 to L.W., HL158081 and CA172408 to F.C.Y., and CA279317 to M.A.B., from National Natural Science Foundation of China 82372736 to L.Z., from the Robert H. Lurie Comprehensive Cancer Center to S.M.L., and from the Canadian Institutes of Health Research to E.B.A. L.W. is an American Cancer Society Research Scholar (RSG-22-039-01-DMC).

## Author contributions

K.S., S.L., C.M., Y.P., S.B., M.K., P.Z., and R.T. performed experiments and together with M.M.Z., T.K., S.M.L., M.A.B., F-C.Y., E.B.A, Z.Z., L.Z., L.W., and T.G.K. analyzed the data. T.G.K. wrote the manuscript with input from all authors.

## Competing interests

The authors declare no competing interests.

## Additional information

[1]Department of Pharmacology, University of Colorado School of Medicine, Aurora, CO, USA. [2]Institute of Translational Medicine, The First Hospital, Jilin University, Changchun, China. [3]International Center of Future Science, Jilin University, Changchun, China. [4]Cell Signaling and Cancer Research Unit, Maisonneuve-Rosemont Hospital Research Center, CIUSSS de l'Est-de-l'Île de Montréal, Montréal, QC, Canada. [5]Department of Cell Systems and Anatomy, Mays Cancer Center, UT Health Science Center San Antonio, San Antonio, TX, USA. [6]Department of Biochemistry and Molecular Genetics, Northwestern University Feinberg School of Medicine, Chicago, IL, USA. [7]Department of Pharmacological Sciences, Icahn School of Medicine at Mount Sinai, New York, NY, USA. [8]Institute of Biomedical Research and Innovation, Foundation for Biomedical Research and Innovation at Kobe, Kobe, Japan. [9]Department of Biomedical Sciences, University of Pennsylvania, School of Veterinary Medicine, Philadelphia, PA, USA. [10]Department of Medicine, University of Montréal, Montréal, QC, Canada. ✉e-mail: leizeng@jlu.edu.cn; lu.wang1@northwestern.edu; tatiana.kutateladze@cuanschutz.edu

