## [Transparent Peer Review file · Nature Communications]

Recruitment of BRD4 to the ASXL1 genomic targets depends on the extra-terminal domain of BRD4

Corresponding Author: Professor Tatiana Kutateladze

Version 0:

Reviewer comments:

Reviewer #1

(Remarks to the Author)

This manuscript by Selvam et al. investigates the interaction between the chromatin protein and transcriptional regulator BRD4 and ASXL1, a key driver of numerous solid tumors. The authors use biochemical and structural data to define the interaction surface between BRD4 and ASXL1, which they map to the ET domain of BRD4 and what is defined as the EBM domain of ASXL1. The ET domain of BRD4 is also shown to bind DNA, though through a distinct interface from ASXL1 binding. The authors demonstrate that BRD4 can similarly bind to the related proteins ASXL2 and 3. Furthermore, to investigate the role for this interaction in transcriptional regulation, chip-seq data shows that both BRD4 and ASXL1 occupy active regions of chromatin, with the majority of ASXL1 localized to promoter regions. In multiple cancer types, ASXL1 and BRD4 expression is elevated and frequently correlated, and the other genes with which each are correlated share highly similar functions. Importantly, the authors also investigate pathological truncations observed for ASXL1 in myeloid malignancies and characterize their binding to BRD4, showing that they do still bind BRD4 and localize to chromatin, although the chromatin binding is independent of BRD4. The truncation mutants are differentiated in their ability to link BRD4 to chromatin, as one mutant shows decreased BRD4 binding relative to the other at key gene promoters. Overall, this manuscript identifies a novel interaction surface between important transcriptional regulators, BRD4 and ASXL1, and nicely combines rigorous biochemical and structural studies with cell-based assays to determine whether these proteins function together in vivo and in cancer cells. The data presented are sound and well-controlled and the interpretations of the data are appropriate. Altogether, the integration of multiple data types to investigate both the biochemical properties and potential shared roles in transcriptional regulation in solid tumors of the BRD4-ASXL1 interaction and the new insights this will provide to a new and important protein-protein interaction interface associated with relevant drug targets indicate that this paper will be of significant interest to a broad range of structural biologists, cell biologists, and cancer biologists, among others.

Below are some relatively minor concerns regarding some of the data presentation in the manuscript, outlined below:

1. It would be useful if the authors could describe in the text why the 568-581 peptide of ASXL1 was selected as the potential interaction surface with BRD4? Previous research of a peptide binding assay is mentioned, however, is this the only peptide that was tested or where other regions of the protein also investigated?
2. It would be helpful to include a domain schematic, or more importantly, an alignment of the EBM sequences of ASXL1, 2, and 3 to better assess the similarity of these proteins.
3. Figure 6c is a useful experiment to demonstrate the binding of BRD4 in the presence of pathological truncation mutants of ASXL1. Are there additional BRD4 target genes that show a similar pattern of binding that could be shown? This would help strengthen the conclusions for this result overall.
4. The graphs in Figure 6 should have some evaluation of the statistical significance of the presented results.
5. Figure 7a is missing a key to indicate the colors of the normal and tumor samples—although this is displayed in a standard format, a legend or description of the colors in the figure legend should be provided.

Reviewer #2

(Remarks to the Author)

Overall, the work by Selvam et al characterises a previously reported interaction between two transcriptional regulators, the ASXL proteins and BRD4. This interaction appears to be exacerbated in the context of truncation variants of ASXL proteins that frequently occur in cancers. The authors use a good combination of biochemistry, structural biology, cell culture and bioinformatics to reach a conclusion that this interaction leads to co-localisation of ASXL1-BRD4 at specific promoter regions. The biochemistry and structural biology is solid and clearly well-executed, providing an experimental structure in line with previous predictions and binding studies. The functional link between overexpression studies analysed here and bioinformatic analyses are interesting, but could be improved by better validation of protein expression levels relative to each other and endogenous protein levels. The simultaneous DNA-EBM binding by BRD4 ET domain are particularly intriguing, but not really expanded on much, even in the discussion, which is a bit unfortunate.

Main Points

The first half of the manuscript is largely an expansion of previously reported studies showing that ASXL truncation generates a gain of function for BRD4 binding (Yang et al, the authors themselves in Szczepanski) and occurs due to an ET-binding motif in ASXL proteins (Burgess et al). The authors go on to further verify this binding, and experimentally determine the structure using NMR. The ET domain binding mechanism, conservation across different ASXL proteins, are mutants identified are generally consistent with previously published results.

The argument is made that this work is validating previous in vitro cell culture studies in a more relevant system. However, most of the experiments are over expressing ASXL1 variants in (mostly HEK293) cell lines. While valid experiments, this does not really seem to be a lot more biologically relevant relative to previous studies. While the detailed ChIPseq adds detail relative to the previous knock-in mouse studies by Yang et al, it is still a bit unclear the role of overexpression in these studies.

The initial ChIP-Seq in Figure 1d – the authors only reference in the methods where their combined datasets are obtained. There is a general lack of information, such as whether these are compared to a mock transfected, IgG control? Presumably these are FL ASXL1, any overexpression.....? Etc

The DNA binding by the ET domain is novel and intriguing. However it is not expanded on much when it could be quite exciting. The gel shift is suggested to show no-effect on binding between DNA and EBM but is relatively qualitative. To make the most of this novel finding, can the NMR-based system, or ITC be used to quantitatively say that EBM-binding does or does not affect DNA binding, or vice versa?

When comparing overexpression (of ASXL1 mutants) to endogenous WT. The authors fail to show whether the overexpression is comparable to an endogenous level or to each other. This applies to the qPCR work also. If the mutations were overexpressed, the FL-WT should have been as well, and the protein normalised to this. Otherwise the difference in promoter occupation, could be as a result of overexpression rather than being relevant.

As both BRD4 and ASXL1 are known transcriptional regulators it would not be unheard of that they would have overlapping ontology terms in the LGG patient analysis? Consider revising to emphasise what is most novel and/or relevant to the mechanism from this data?

Minor points

The structural figures are generally over-labelled, to the point where the info that is being conveyed is obscured. Suggestion would be to only point out key residues. Consider pop out boxes to highlight specific residue interactions. Consider using a different colour to represent ASXL1 as it is not color-blind accessible on some coloured surfaces.

Figure 3e-f could be clarified with better colour scheme and more judicious labelling.

There is no explanation as to why the 601 DNA was chosen- highlight the rationale for this in text.

Figure 6 appears to use SEM with two samples. Is this statistically valid?

Figure 7 When looking at the expression in cancerous tissue, the authors report that this was compared to "normal tissue" but do not provide details of this. Nor do they define what LGG patients are. There is also no key as to what the colours in these graphs mean.

Suppl figure 3- illustrates one of the main novel findings in the paper. I would suggest moving This to the main text.

Lack of in text referencing: Supplementary table is not referenced in the text, nor Suppl. Figure 3.

The Methods section lacks sufficient detail. For cell culture, include the growth conditions, particularly in the ChIP-seq section. Details on how the purifications were performed are also limited—review these procedures and add enough information to allow replication. Methods are also lacking for the mass spec experiments.

Reviewer #3

(Remarks to the Author)

Reviewer #4

(Remarks to the Author)

Selvam et al. report structural and genomic data linking BRD4 and ASXL1 in the regulation of transcription and cancer. The results are interesting but mostly confirmatory or incremental over previous publications on the same subject. Burgess et al. (reference 26) already characterized the interaction, measured the affinity of the ET domain of BRD4 for the motifs of ASXL1, 2, and 3, and mapped the binding site by mutagenesis. Selvam et al. mention that their affinity measurement is similar, but the general agreement with this previous characterization is not properly discussed. Selvam et al. have experimentally determined the structure of the BRD4-ASXL1 complex, which seems to be highly similar to that of BRD4-NSD3 (reference 21), as well as BRD3-CH1 and BRD3-BRG1 (not referenced). This structural similarity is not discussed. The functional link is further supported by genomic data, measured by the authors or analyzed by the authors using public databases. Overall, the results of this work are valuable, but do not provide as much novelty and relevance as the authors think. The situation would be different if they could provide a mechanistic insight into the increased affinity of the truncated forms of ASXL1 for BRD4 and how this gain-of-function relates to cancer.

Version 1:

Reviewer comments:

Reviewer #1

(Remarks to the Author)

The concerns raised in my review have been addressed by the authors. I have no additional issues with this manuscript.

Reviewer #2

(Remarks to the Author)

The authors more thorough method description and additional data (e.g. clarifying of levels for proteins subsequently used for downstream analysis, testing effects on DNA binding etc) have certainly improved the manuscript.

The additional data regarding MLL binding is interesting, although any alphafold modelling that is displayed (Fig 7g) should include reports of error scores and a PAE plot illustrating confidence in relevant interfaces

While the authors argue that labelling is 'standard' I would encourage the editor to make their own judgement about how well labelling in figures like Fig 2c or 3f helps to highlight relevant information.

Reviewer #3

(Remarks to the Author)

Reviewer #4

(Remarks to the Author)

In my view, it will be enriching, not distracting, to mention and reference previous related work.

The authors should revise the structure presented in their work, as there are many inconsistencies between the statistics summarized in Supplementary Table 1 and the table on page 23 of the PDB validation report. Protein residue numbering in the text and figures is also inconsistent with the numbering used in Supplementary Table 1.

Version 2:

Reviewer comments:

Reviewer #4

(Remarks to the Author)

Excuse me if I am wrong, but I have downloaded both the PDB report and the manuscript, and the NMR restraints statistics on page 23 of the report are very different from those in Supplementary Table 1. Snapshots of both are shown in the attached file.

We would like to thank the Reviewers for their insightful comments, which were very helpful in revising and strengthening this manuscript.

Reviewer #1 (Remarks to the Author):

This manuscript by Selvam et al. investigates the interaction between the chromatin protein and transcriptional regulator BRD4 and ASXL1, a key driver of numerous solid tumors. The authors use biochemical and structural data to define the interaction surface between BRD4 and ASXL1, which they map to the ET domain of BRD4 and what is defined as the EBM domain of ASXL1. The ET domain of BRD4 is also shown to bind DNA, though through a distinct interface from ASXL1 binding. The authors demonstrate that BRD4 can similarly bind to the related proteins ASXL2 and 3. Furthermore, to investigate the role for this interaction in transcriptional regulation, chip-seq data shows that both BRD4 and ASXL1 occupy active regions of chromatin, with the majority of ASXL1 localized to promoter regions. In multiple cancer types, ASXL1 and BRD4 expression is elevated and frequently correlated, and the other genes with which each are correlated share highly similar functions. Importantly, the authors also investigate pathological truncations observed for ASXL1 in myeloid malignancies and characterize their binding to BRD4, showing that they do still bind BRD4 and localize to chromatin, although the chromatin binding is independent of BRD4. The truncation mutants are differentiated in their ability to link BRD4 to chromatin, as one mutant shows decreased BRD4 binding relative to the other at key gene promoters. Overall, this manuscript identifies a novel interaction surface between important transcriptional regulators, BRD4 and ASXL1, and nicely combines rigorous biochemical and structural studies with cell-based assays to determine whether these proteins function together in vivo and in cancer cells. The data presented are sound and well-controlled and the interpretations of the data are appropriate. Altogether, the integration of multiple data types to investigate both the biochemical properties and potential shared roles in transcriptional regulation in solid tumors of the BRD4-ASXL1 interaction and the new insights this will provide to a new and important protein-protein interaction interface associated with relevant drug targets indicate that this paper will be of significant interest to a broad range of structural biologists, cell biologists, and cancer biologists, among others.

Below are some relatively minor concerns regarding some of the data presentation in the manuscript, outlined below:

1. It would be useful if the authors could describe in the text why the 568-581 peptide of ASXL1 was selected as the potential interaction surface with BRD4? Previous research of a peptide binding assay is mentioned, however, is this the only peptide that was tested or where other regions of the protein also investigated? – we used the peptide two residues shorter than the peptide from ref. 26 but haven't investigated other regions of ASXL1. We have added discussion on possible roles of other domains in conclusion, on page 14.
2. It would be helpful to include a domain schematic, or more importantly, an alignment of the EBM sequences of ASXL1, 2, and 3 to better assess the similarity of these proteins. – the alignment is now shown in new Suppl. Fig. 2.
3. Figure 6c is a useful experiment to demonstrate the binding of BRD4 in the presence of pathological truncation mutants of ASXL1. Are there additional BRD4 target genes that show a similar pattern of binding that could be shown? This would help strengthen the conclusions for this result overall. – as suggested, additional example is shown in new Suppl. Fig. 6.
4. The graphs in Figure 6 should have some evaluation of the statistical significance of the presented results. – we have added statistical analysis.
5. Figure 7a is missing a key to indicate the colors of the normal and tumor samples—although this is displayed in a standard format, a legend or description of the colors in the figure legend should be provided. – we have added color key in (now Fig. 8a) and defined colors in the legend.

Reviewer #2 (Remarks to the Author):

Overall, the work by Selvam et al characterises a previously reported interaction between two transcriptional regulators, the ASXL proteins and BRD4. This interaction appears to be exacerbated in the context of truncation variants of ASXL proteins that frequently occur in cancers. The authors use a good combination of biochemistry, structural biology, cell culture and bioinformatics to reach a conclusion that this interaction leads to co-localisation of ASXL1-BRD4 at specific promoter regions. The biochemistry and structural biology is solid and clearly well-executed, providing an experimental structure in line with previous predictions and binding studies. The functional link between overexpression studies analysed here and bioinformatic analyses are interesting, but could be improved by better validation of protein expression levels relative to each other and endogenous protein levels. The simultaneous DNA-EBM binding by BRD4 ET domain are particularly intriguing, but not really expanded on much, even in the discussion, which is a bit unfortunate.

Main Points

The first half of the manuscript is largely an expansion of previously reported studies showing that ASXL truncation generates a gain of function for BRD4 binding (Yang et al, the authors themselves in Szczepanski) and occurs due to an ET-binding motif in ASXL proteins (Burgess et al). The authors go on to further verify this binding, and experimentally determine the structure using NMR. The ET domain binding mechanism, conservation across different ASXL proteins, are mutants identified are generally consistent with previously published results.

The argument is made that this work is validating previous in vitro cell culture studies in a more relevant system. However, most of the experiments are over expressing ASXL1 variants in (mostly HEK293) cell lines. While valid experiments, this does not really seem to be a lot more biologically relevant relative to previous studies. While the detailed ChIPseq adds detail relative to the previous knock-in mouse studies by Yang et al, it is still a bit unclear the role of overexpression in these studies.

– the major point of this work is to understand the molecular-level mechanism of the recognition of ASXL1 by BRD4 (which is essential for pharmacological targeting, especially taking into account the significance of both proteins in oncogenesis) and to uncover the direct effect of this interaction by comparing wt and loss-of-function ASXL1 in biological assays (both endogenous proteins (Figs. 1d,e, 6d, 7) and stably expressed proteins (Fig. 5) were investigated). Please note that we do not investigate ‘that ASXL1 truncation generates a gain of function’. This fact is mentioned only in concluding remarks (page 14): “...the cancer-specific truncation variants of ASXL1 but not full-length ASXL1 have previously been shown to appreciably interact with BRD4 and induce myeloid malignancies in animal models^{6,26,35}.”

Our finding regarding the difference in BRD4 colocalization with the longer ASXL1(1-645) and shorter ASXL1(1-591) variants prompted us to test the association of these variants with the methyltransferases MLL3/4, which recognize aa 607-627 of ASXL1. As shown in new Fig. 7, while the shorter variant is lacking MLL3/4 binding sequence and is incapable of binding to MLL3/4, the longer construct binds both BRD4 and MLL3/4 concurrently and independently. We discuss possible implications of such a difference in concluding remarks.

The initial ChIP-Seq in Figure 1d – the authors only reference in the methods where their combined datasets are obtained. There is a general lack of information, such as whether these are compared to a mock transfected, IgG control? Presumably these are FL ASXL1, any overexpression.....?

– we have expanded the text describing data shown in Fig. 1d (and Fig. 1d legend) to clarify this: page 5: “...we performed chromatin immunoprecipitation coupled with deep sequencing (ChIP-seq) of endogenous ASXL1, BRD4 and BAP1 and assessed genomic occupancies of these proteins in human HEK293T

cells. We also analyzed ChIP-seq datasets of histone H3 modifications, including H3K4me3, H3K4me1, H3K27ac and H3K27me3, previously generated by us from human HEK293T cells ²⁷.”

page 29 (legend): “Heat maps of ChIP-seq signals of the indicated endogenous full length proteins and PTMs at genomic regions bound by BRD4 in HEK293T cells.”

Fig. 1d shows heat maps, therefore there is no mock transfection/IgG to compare to. For other Figures where peaks were called, the peaks were called relative to GFP ChIPs in cells transfected with GFP (this information is now provided in the methods section).

page 25 (data availability): “The genomics data for endogenous full-length ASXL1, BRD4 and BAP1... generated in this study have been deposited.... Another dataset previously generated by us and used in this study (histone H3 PTMs in HEK293T cells) is available under accession....”

The DNA binding by the ET domain is novel and intriguing. However it is not expanded on much when it could be quite exciting. The gel shift is suggested to show no-effect on binding between DNA and EBM but is relatively qualitative. To make the most of this novel finding, can the NMR-based system, or ITC be used to quantitatively say that EBM-binding does or does not affect DNA binding, or vice versa? – as suggested, we have measured binding affinity of BRD4_{ET} to the ASXL1 peptide in the presence of DNA to confirm that the DNA binding does not affect ASXL1 binding (new Suppl. Fig. 4). We also show that BRD4_{ET} does not aid in the association of bromodomains of BRD4 with RNAs (new Fig. 4h, i).

When comparing overexpression (of ASXL1 mutants) to endogenous WT. The authors fail to show whether the overexpression is comparable to an endogenous level or to each other. This applies to the qPCR work also. If the mutations were overexpressed, the FL-WT should have been as well, and the protein normalised to this.

Otherwise the difference in promotor occupation, could be as a result of overexpression rather than being relevant. – please note that we do not compare overexpressed ASXL1 variants to endogenous WT ASXL1 in IP, WB or ChIP-qPCR assays in this work (the label WT denotes truncations without mutations). [and as FL ASXL1 is known to be unstable (Inoue, JCI, 2013; Wang, Nat Cancer, 2021), normalization to the FL protein would lead to considerable errors.] We normalize all IP-WB samples to each other using Nanodrop

concentration measurements (please see methods and new Suppl. Fig. 5). We have also added WB of ChIP-qPCR samples that confirms equal expression of GFP-ASXL1 truncation constructs in the Source Data file. Lastly, we use the percent of input, a common method for ChIP-qPCR data normalization and included IgG as a negative control.

As both BRD4 and ASXL1 are known transcriptional regulators it would not be unheard of that they would have overlapping ontology terms in the LGG patient analysis? Consider revising to emphasise what is most novel and/or relevant to the mechanism from this data? – the overlap could be expected, but it is not always the case. Our analysis of BRD4 with other subunits of the BAP1/ASXL1 complex did not show as strong correlation in LGG (no correlation with BAP1, see Figure above), making the BRD4-ASXL1 association unique and novel in this cancer. We note that analysis shown in Fig. 8 has not been done previously and is novel, as well as the finding that BRD4 occupies the ASXL1 promoter, especially in glioblastoma. We further propose the feed-forward mechanism ‘that would explain the high correlation of ASXL1 and BRD4 in malignancies, particularly in glioblastoma’ (pages 12, 14 and 3 (abstract)).

Minor points

The structural figures are generally over-labelled, to the point where the info that is being conveyed is obscured. Suggestion would be to only point out key residues. Consider pop out boxes to highlight specific residue interactions. Consider using a different colour to represent ASXL1 as it is not color-blind accessible on some coloured surfaces. Figure 3e-f could be clarified with better colour scheme and more judicious labelling. – Labeling in crystallographic and NMR data Figures is relatively standard and is required to indicate all detected hydrogen bonding/electrostatic/hydrophobic contacts and CSPs.

There is no explanation as to why the 601 DNA was chosen- highlight the rationale for this in text. – Widom 601 DNA is one of the most common reagents in chromatin biology.

Figure 6 appears to use SEM with two samples. Is this statistically valid? – we have corrected Fig. 6d to show four experiments, thank you for pointing to this.

Figure 7 When looking at the expression in cancerous tissue, the authors report that this was compared to “normal tissue” but do not provide details of this. Nor do they define what LGG patients are. There is also no key as to what the colours in these graphs mean. – the methods section ‘Tumor vs normal expression profiling’ has been expanded to include normal tissues definition and numbers of samples for tumor and normal tissues for each lineage. We have added color key in (now Fig. 8a) and defined colors in the legend. LGG is defined in Fig. 8a legend.

Suppl figure 3- illustrates one of the main novel findings in the paper. I would suggest moving This to the main text. – this image is now shown in Fig. 4d.

Lack of in text referencing: Supplementary table is not referenced in the text, nor Suppl. Figure 3. – corrected both.

The Methods section lacks sufficient detail. For cell culture, include the growth conditions, particularly in the ChIP-seq section. Details on how the purifications were performed are also limited—review these procedures and add enough information to allow replication. Methods are also lacking for the mass spec experiments. – we have expanded the methods section accordingly.

Reviewer #3 (Remarks to the Author):

Reviewer #4 (Remarks to the Author):

Selvam et al. report structural and genomic data linking BRD4 and ASXL1 in the regulation of transcription and cancer. The results are interesting but mostly confirmatory or incremental over previous publications on the same subject. Burgess et al. (reference 26) already characterized the interaction, measured the affinity of the ET domain of BRD4 for the motifs of ASXL1, 2, and 3, and mapped the binding site by mutagenesis. Selvam et al. mention that their affinity measurement is similar, but the general agreement with this previous characterization is not properly discussed. Selvam et al. have experimentally determined the structure of the BRD4-ASXL1 complex, which seems to be highly similar to that of BRD4-NSD3 (reference 21), as well as BRD3-CH1 and BRD3-BRG1 (not referenced). This structural similarity is not discussed. The functional link is further supported by genomic data, measured by the authors or analyzed by the authors using public databases. Overall, the results of this work are valuable, but do not provide as much novelty and relevance as the authors think. The situation would be different if they could provide a mechanistic insight into the increased affinity of the truncated forms of ASXL1 for BRD4 and how this gain-of-function relates to cancer.

– It would be very interesting to resolve the question as to why FL ASXL1 is unstable, whereas some cancer truncated variants are stable and bind to BRD4 more tightly, however it's an enormously challenging task to accomplish using currently available structural biology tools. Not only FL ASXL1 is unstable, it also contains a ~1000 aa disordered [predicted] region between the N-terminal domains and the C-terminal PHD/Zn finger, which could undergo proteolytic cleavage (explaining the instability) and serve as binding sites for co-factors, for example BRD4 and MLL3/4 (we have added new data regarding MLL3/4 interactions in Fig. 7).

Burgess et al, ref. 26; it's an inspiring work, we confirmed K_{dS} ["The dissociation constant (K_d) was found to be 1.6 μ M, and was in agreement with the previously reported affinity of 1.2 μ M²⁶"], cite this paper four times, including in Introduction: "A region in ASXL1 has recently been identified as a binding partner of the ET domain of BRD4²⁶, prompting us to explore the mechanistic details and the functional significance of this interaction." And the first sentence of results states: "Binding of BRD4_{ET} to an ASXL1 peptide was originally observed in biochemical pulldown experiments²⁶."

For comparing to other ET domains structures. We usually do such a comparison for structure/biochemical studies that do not involve biological research, and this analysis is straightforward and easy to do. However, for studies focusing on a specific interaction and biological and functional implications of this interaction, such comparison [to the complexes with unrelated ligands or homologous proteins] would be perceived as a distraction from, rather than help in understanding the major point.

We would like to thank the Reviewers for their insightful comments, which were very helpful in revising and strengthening this manuscript.

Reviewer #2:

The authors more thorough method description and additional data (e.g. clarifying of levels for proteins subsequently used for downstream analysis, testing effects on DNA binding etc) have certainly improved the manuscript.

The additional data regarding MLL binding is interesting, although any alphafold modelling that is displayed (Fig 7g) should include reports of error scores and a PAE plot illustrating confidence in relevant interfaces. – we appreciate this comment, though we did not run the AF modeling, it is available in UniProt. We have added to the Fig. 7g legend that the AF model is taken 'from UniProt, #Q8IXJ9'.

[... While the authors argue that labelling is 'standard' I would encourage the editor to make their own judgement about how well labelling in figures like Fig 2c or 3f helps to highlight relevant information. – in the first round of reviews this reviewer inadvertently suggested to cherry pick (highlight) some, rather than all, experimentally observed contacts/perturbations, i.e. show incomplete analysis in figures]

Reviewer #4:

In my view, it will be enriching, not distracting, to mention and reference previous related work. – as suggested, in addition to citing structures of BRD4, we have added reference to the structures of BRD3-CHD4 and BRD3-BRG1 (ref. #28).

The authors should revise the structure presented in their work, as there are many inconsistencies between the statistics summarized in Supplementary Table 1 and the table on page 23 of the PDB validation report. – the single difference is the number of hydrogen bonds. We have updated Suppl. Table 1 (revised this number from 44 to 50 (two restraints per one hydrogen bonds=100 restraints), as it is shown on page 23 of the validation report).

Protein residue numbering in the text and figures is also inconsistent with the numbering used in Supplementary Table 1. – the numbering of BRD4 and ASXL1 in Suppl. Table 1 has been updated.

We would like to thank the Reviewers for their insightful comments, which were very helpful in revising and strengthening this manuscript.

Reviewer #4:

Excuse me if I am wrong, but I have downloaded both the PDB report and the manuscript, and the NMR restraints statistics on page 23 of the report are very different from those in Supplementary Table 1. Snapshots of both are shown in the attached file. – we very much appreciate this comment and sincerely apologize for the confusion. The corrected validation report, where statistics on page 23 match statistics in the Suppl. Table 1, is now included.